# Dynamically cultured, differentiated bovine adipose-derived stem cell spheroids as building blocks for biofabricating cultured fat

Annemarie Klatt[1,4], Jannis O. Wollschlaeger[1,4], Franziska B. Albrecht[1], Sara Rühle[2], Lena B. Holzwarth[2], Holger Hrenn[3], Tanja Melzer[3], Simon Heine[1] & Petra J. Kluger[2]

Cultured or cultivated meat, animal muscle, and fat tissue grown in vitro, could transform the global meat market, reducing animal suffering while using fewer resources than traditional meat production and no antimicrobials at all. To ensure the appeal of cultured meat to future customers, cultured fat is essential for achieving desired mouthfeel, taste, and texture, especially in beef. In this work we show the establishment of primary bovine adipose-derived stem cell spheroids in static and dynamic suspension culture. Spheroids are successfully differentiated using a single-step protocol. Differentiated spheroids from dynamic cultures maintain stability and viability during 3D bioprinting in edible gellan gum. Also, the fatty acid composition of differentiated spheroids is significantly different from control spheroids. The cells are cultured antibiotic-free to minimize the use of harmful substances. This work presents a stable and bioprintable building block for cultured fat with a high cell density in a 3D dynamic cell culture system.

The research field of producing cultivated, or cultured meat (CM) in vitro has been growing rapidly within the last few years. In 2022, USD 896 million were invested in CM and seafood companies worldwide, bringing the all-time investment up to USD 2.8 billion[1]. The growing investment is fueled by the trendsetting prospects of CM. In the future, it could contribute to the nutrition of a growing world population while producing fewer greenhouse gas emissions and using less land than conventional animal husbandry[2,3]. The impact of meat production on the environment, as well as the overall use of antimicrobials in food production, could be reduced and animal welfare would also improve[4–6]. To live up to those claims, CM research and development needs to conquer many hurdles in all parts of the process[7,8]. Finding a cell source suitable for mass production, culturing cells without animal components like fetal bovine serum (FBS), the scale-up of cell mass,

the structure of a whole meat cut, the regulatory approval, taste, texture, and composition of the product and consumer acceptance are critical points in CM development[9]. Therefore, just as in a classic meat product, (cultured) fat is especially critical for the mouthfeel, texture, taste, and with that, customer acceptance of the final product[10–16]. The fat content and distribution are essential in the perception and rating of the quality of a meat cut, especially in beef[17]. In vivo, adipose tissue, or fat, develops via adipogenesis when mesenchymal stem cells (MSCs) differentiate into pre-adipocytes, which will further undergo terminal differentiation. During this process, cells accumulate lipids and subsequently enlarge and form univacuolar mature adipocytes[18]. To produce cultured adipocytes, the process of adipogenesis can be induced through a mixture of different supplements in the culture medium. Some of the classical differentiation supplements used

[1]Reutlingen Research Institute, Reutlingen University, Reutlingen, Germany. [2]Faculty of Life Sciences, Reutlingen University, Reutlingen, Germany. [3]Core Facility Hohenheim, University of Hohenheim, Stuttgart, Germany. [4]These authors contributed equally: Annemarie Klatt, Jannis O. Wollschlaeger.
✉ e-mail: petra.kluger@reutlingen-university.de

in vitro are insulin, dexamethasone, hydrocortisone, 3-isobutyl-1-methylxanthine (IBMX), indomethacin and rosiglitazone[19,20]. Briefly summarized, those supplements induce the adipogenesis by targeting either directly or indirectly the key driver of adipogenesis: the transcription factor peroxisome proliferator-activated receptor gamma (PPARγ). IBMX and glucocorticoids like dexamethasone and hydrocortisone initiate the transcription of PPARγ via CCAAT-enhancer-binding proteins beta/delta[21–23]. Insulin activates the transcription of PPARγ via a signal cascade. While glucocorticoids and the anti-inflammatory drug indomethacin activate PPARγ directly, IBMX can activate it indirectly[24,25]. Rosiglitazone, an insulin sensitizer, binds and activates PPARγ and makes the cells more responsive to insulin, enhancing its effect on the cells[26]. Usually, adherent cells, including bovine adipose-derived stem cells (bASCs), are cultured and differentiated in a 2D cell culture. This culture technique is time, work and material consuming, making cell mass production costly[27,28]. Hence culturing cells in a preferably dynamic 3D cell culture system is mandatory for large-scale CM production[29,30]. When employing non-edible scaffolds such as microcarriers, an additional step is necessary to harvest the cells before proceeding with further processing[31]. To avoid this extra step, edible microcarriers can be used to culture the cells in a dynamic system[32,33]. Another approach for dynamic 3D tissue culture does not require any scaffold. Here, cells aggregate and form spheroids[34,35]. In CM production, a high cell density in the final product is necessary to achieve a product comparable to conventional meat. In spheroids, cells are packed densely and in combination with additional edible non-animal materials (such as alginate, gellan gum (GG), etc.), they enable the formation of larger structures. Thereby, spheroids facilitate manufacturing on an industrial scale[36]. Therefore, spheroids have the potential to function as a convenient, simple, and excellent building block in CM research. Very little research has been conducted concerning bovine fat spheroids and fat spheroids of other species[37,38]. In this study, we differentiate, print, and analyze primary bASC spheroids after dynamic culture to showcase the potential of bASC spheroids in CM research and production. We demonstrate antibiotic-free culture of static and dynamic bASC spheroids, their differentiation, 3D extrusion-based bioprinting in edible GG, and analysis of the fatty acid composition.

## Results

### Challenges in cultured meat commercialization and our pathways to solutions

The development of a dynamic spheroid culture of bASCs for CM is a delicate process that does not only need a stable foundation of knowledge concerning isolation, culture and differentiation of bASCs. It also needs to address the challenges of CM research and production. Figure 1a gives a simplified overview of the development pathways for CM, including some of the most important technological, financial, consumer, and regulatory gaps that need to be addressed for CM commercialization. In our study, we have addressed key aspects of these development pathways, presenting a clear roadmap for the efficient and scalable production of cultured fat (Fig. 1b).

### The one-step differentiation protocol was optimized

As a first step towards establishing a stable spheroid culture, bASCs need to be isolated and characterized, while the differentiation protocol needs to be simplified and optimized. The enzymatic isolation of bASCs from slaughter waste of adult Limousin cattle yielded about 283,500 (±182,000) cells per gram of fat tissue (Fig. 2a). The number of isolated cells was highly donor-dependent and varied between the subcutaneous fat for the different body regions (Supplementary Table 1). When determining the growth kinetics, the doubling time in the exponential phase was found to be 25.73 (±3.59) h (Fig. 2b). Isolated cells displayed a fibroblast-like morphology and were analyzed for the expression of different CD markers to verify cell identity. The

cells were found positive for the marker proteins CD73, CD90, and CD105 while lacking the expression of CD56 (Fig. 2c). Various differentiation protocols were used to determine the adipogenic differentiation capacity of the cells. In the process, the differentiation protocol was adapted to be as simple as possible while maintaining a high differentiation efficiency. All protocols were able to differentiate the cells in a single step over the course of 14 days. The testing revealed that indomethacin induced differentiation in bASCs but it induced visually less lipid accumulation compared to rosiglitazone (Figs. 2d, e and Supplementary Figs. 1a, b). Within the bASCs, small lipid-filled vacuoles formed over time, already visible at day 7 (Fig. 2f and Supplementary Fig. 1c). After 14 days in differentiation medium containing IBMX, rosiglitazone, insulin and dexamethasone, the cells accumulated lipids. The cells displayed a morphology different from the characteristic spindle-shaped fibroblast one of ASCs. An extended differentiation period of 21 days did not significantly improve the lipid accumulation (Fig. 2f and Supplementary Fig. 1d). Also, IBMX was not found to increase the lipid accumulation when present in the differentiation medium (Fig. 2g). The final differentiation medium (DMEM-Diff) was based on the proliferation medium with only three supplements: 2.5 μM rosiglitazone, 3 μg/mL bovine insulin and 1 μM dexamethasone.

### A static spheroid culture of primary bASCs was established

After the successful isolation and characterization of primary bASCs, a static 3D spheroid culture was established. Spheroid formation under static conditions took place within two days using the liquid overlay method. Thereby, the cells are applied in a layer of culture medium over a non-adhesive surface as described in the "methods" section. Overall, the spheroids displayed an even, round shape with sharp and smooth edges. The spheroids proved to be stable in static culture 14 days after formation (Supplementary Fig. 2a). Within the first 5 days of culture, the volume and diameter of the spheroids decreased. Interestingly, the diameter of spheroids with a starter cell count of 50,000 cells remained above 400 μm, which can be considered above the diffusion limit[37] (Fig. 3a and Supplementary Fig. 2b-c).

Therefore, the viability of cells within the spheroids was investigated. After 14 days in culture, different cell death markers were evaluated. Specific markers for apoptosis (cleaved caspase-3, cCas3), necroptosis (receptor interacting serine/threonine kinase 3, RIPK3), and hypoxia (hypoxia-inducible factor 1, alpha subunit, Hif-1 α) were neither present in spheroids after 14 days of culture nor in spheroids after 14 days in differentiation medium (Fig. 3b). Differentiated spheroids had a larger diameter than the proliferation control on day 14. PPARγ was not found in the control spheroids but was widely expressed in the differentiated spheroids (Fig. 3c). Also, the expression and even distribution of perilipin 1, a lipid-droplet associated protection protein, was verified around the cell vacuoles (Fig. 3d). Within the spheroids of the control group, very little lipid accumulation was found. The lipid staining revealed that differentiated bASC spheroids accumulated lipids on the outside of the spheroid as well as on the inside (Fig. 3e). The semi-quantitative assessment of the two differentiation markers (PPARγ, perilipin 1) and the lipid staining revealed significantly increased fluorescence intensities on day 14 of differentiation compared to the proliferation control on day 14 (Supplementary Fig. 2d).

### bASC spheroids can be cultured/differentiated dynamically

Upscaling is one of the main concerns in CM research and development. Therefore, a dynamic suspension culture of bASC spheroids was established using orbital shakers. Bovine ASC spheroids in dynamic suspension culture formed within two days. Spheroids were formed in a range of different sizes within a culture dish. The size of the spheroids was influenced by the shaking speed and culture density. The size distribution was most uniform when maintaining a culture density of

## a) Product Development Pathways for Cultured Meat

**Technology**
- Cell growth
- Differentiation [2]
- No Antibiotics [1]
- No Serum
- 3D structure [5]
- Texture

**Financial**
- Scalable culture [3]
- Suspension culture [3]
- High cell density [4]
- Short culture time [2]
- Fewer disposables [3]
- Cheap media

**Consumer**
- Availabilty
- Taste/Aroma
- Appearance
- Mouthfeel
- Marketing
- Designation

**Regulation**
- Legal framework
- Process/SOPs
- Composition [6]
- Hazards/risks
- Risk assassement
- Labelling

## b) Engineering Cultured Fat

[1] Antibiotic-free media

[3] Suspension culture/ no microcarriers

[5] Edible and 3D-printable materials

[2] Simple one-step differentiation

[4] Cultured meat building blocks

[6] Fatty acid profile close to native fat

**Fig. 1 | Hurdles and technological barriers for CM commercialization. a** Product development pathways for CM showing the technological, financial, consumer, and regulatory gaps that need to be addressed for successful CM commercialization. **b** Schematic overview of the work process in relation to the product development pathways for CM. First, creation of a one-step differentiation protocol without the use of antibiotics and then the establishment of static and dynamic primary bASC spheroids. This is followed by the characterization, bioprinting and fatty acid analysis of the spheroids.

250,000 cells/mL at a shaking speed of 70 rpm during spheroid formation. For spheroid maintenance and differentiation, a shaking speed of 80 rpm led to the most favorable size distribution of the spheroids while preserving cell viability. Spheroid morphology was oval to round and with sharp borders and not as uniform as seen within the static culture (Fig. 4a). The spheroids could easily be harvested after culture to determine the weight of the final cell mass relative to the starting cell number. To acquire 3.68 mg of biomass after dynamic suspension culture, 1 million cells were needed as starting cell number ($n = 4$; standard deviation $\pm 1.9$ mg/$1 \times 10^6$ cells). Using the same differentiation protocol as used in 2D cell culture, the cells were differentiated for 14 days. After the end of the culture or differentiation period, the largest spheroids were selected to be analyzed for possible cell death in the spheroid core (Fig. 4b). Comparable to the spheroids in static culture, none of the three cell death markers were found within the spheroids. The PPARγ expression within the dynamic spheroids was found evenly distributed in differentiated spheroid on day 14 while none could be found in the control spheroids (Fig. 4c). Also, cells expressing perilipin 1 were found evenly distributed within the differentiated spheroids and could not be found in the undifferentiated control group (Fig. 4d). Additionally, differentiated spheroids presented evenly distributed cells with lipid accumulation within the

spheroids after 14 days in differentiation medium (Fig. 4e). Lipid accumulation was not limited to the areas directly exposed to the medium but was also visible in the spheroid core. Spheroids cultured in control medium showed minimal to no lipid accumulation. The semi-quantitative analyzes of the fluorescence intensities of PPARγ, perilipin 1, and the lipid staining showed significantly higher values on day 14 of differentiation compared to the proliferation control at the same time point (Supplementary Fig. 3a).

### bASC spheroids are suitable for 3D bioprinting

Addressing the suitability of bASC spheroids for the fabrication of CM products, we decided to investigate the printability of the spheroids in a GG-based bioink. The shear-thinning behavior of the bioink becomes apparent when its viscous and elastic modulus is analyzed and quantified. Up to a temperature of ~37 °C, the elastic modulus was higher than the viscous modulus indicating gelation and, therefore, a better print stability and fidelity (Fig. 5a). Further, the inversion test supported the viscous and elastic modulus findings. As the bioink flowed down at 37 °C and gelated at room temperature (23 °C). The resolution of the bioprinted macroscopic grid (1.5% GG) with and without spheroids (Fig. 5b) demonstrated not only an acceptable print fidelity but also a regular distribution of the spheroids throughout the whole

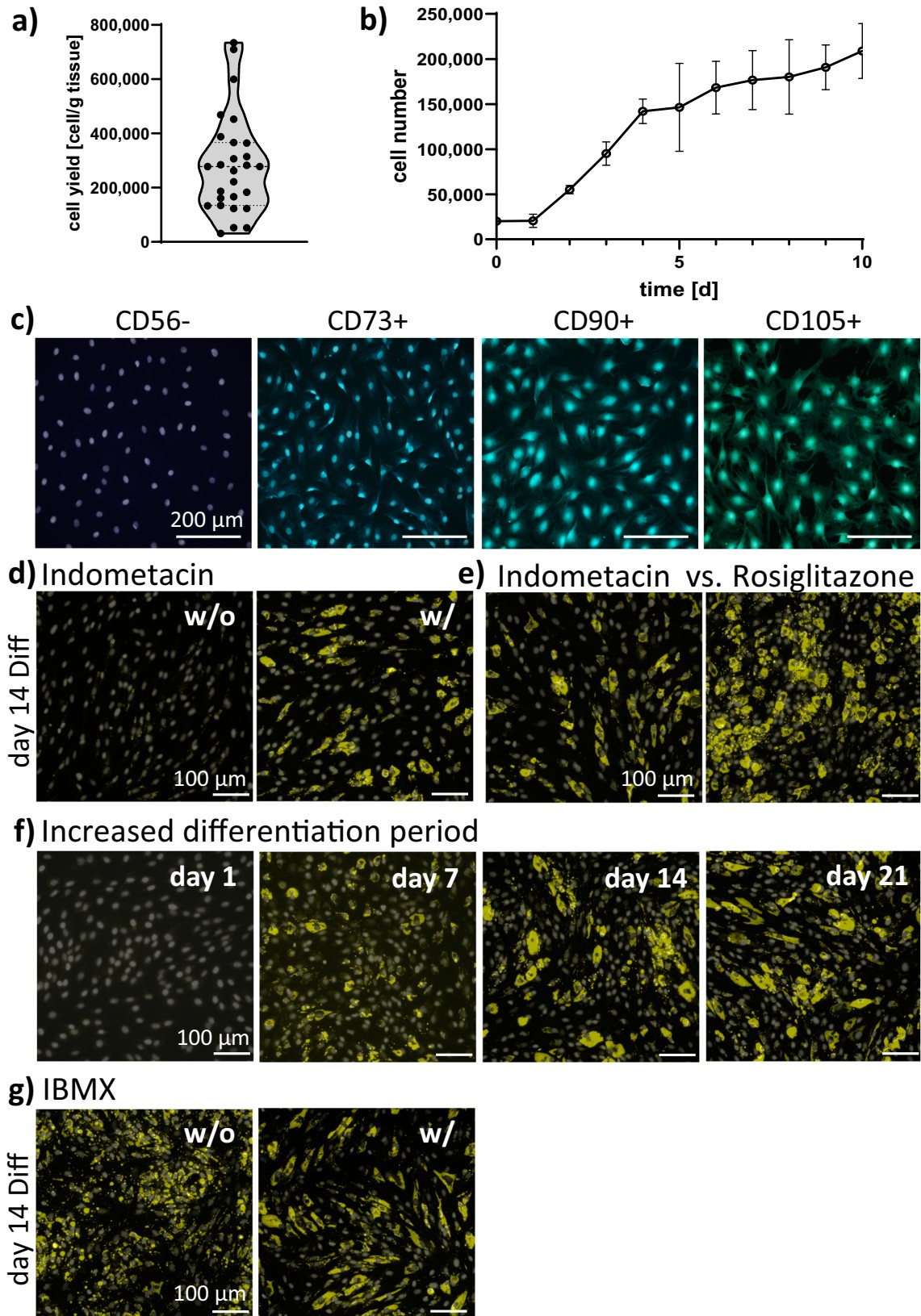

bioink. Spheroids appeared as white dots within the bioprinted grid. Although the spheroids were varying in size, they were quite evenly spread within the printed constructs (Fig. 5b and Supplementary Fig. 3b). Cellular analysis revealed that the spheroids exhibited high cell viability immediately (1 h) after bioprinting and after three days of culture. While most cells within the spheroids were viable, few cells of

the spheroids showed ethidium homodimer-1 staining, indicating some cell death (Fig. 5c). Staining of the intracellular lipids with BODIPY revealed uniform staining across the bioprinted spheroids and, therefore, lipid accumulation across the whole spheroids (Fig. 5d). Spheroid stability and differentiation were not affected by the bioprinting process.

**Fig. 2 | Characterization of bASC. a** Cell yield of bASC primary isolations $n = 27$, violin plot cut at lowest and highest value, dotted line: quartile, dashed line: median, dots represent single data points; **b** Growth kinetics of primary isolated bASCs. Data are plotted as mean and ± standard deviation (SD); **c** Immunofluorescence staining of different CD markers, scale bar: 200 μm; **d** Day 14 of 2D bASC differentiation with (w/) and without (w/o) 100 μM indomethacin. DMEM-Diff (DMEM hg, 2.5 μM rosiglitazone, 3 μg/mL bovine insulin and 1 μM

dexamethasone) - rosiglitazone + 500 μM 3-Isobutyl-1-methylxanthin (IBMX), scale bar: 100 μm; **e** Day 14 of 2D bASC differentiation indomethacin vs rosiglitazone. DMEM-Diff + IBMX; scale bar: 100 μm; **f** 2D bASC differentiation over time until day 21. DMEM-Diff + IBMX, scale bar: 100 μm; **g** Day 14 of 2D bASC differentiation w/ and w/o IBMX present in DMEM-Diff, scale bar: 100 μm; CD markers: blue/green, lipids stained with BODIPY: yellow, cell nuclei: white.; $n = 3$ biological replicates. Source data are provided as a Source Data file.

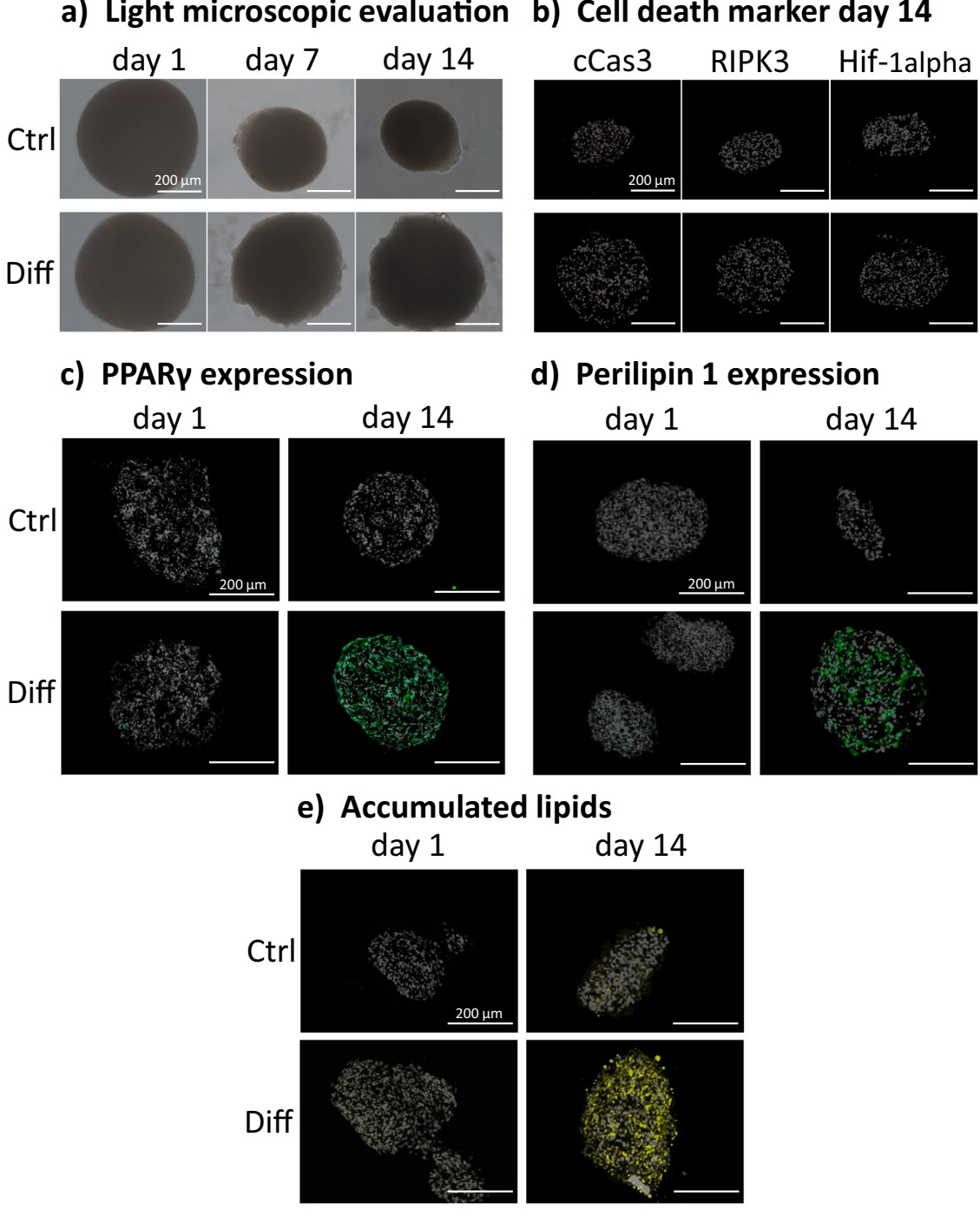

**Fig. 3 | Evaluation of static primary bASC spheroid culture with 50,000 cells during maintenance (Ctrl) and adipogenic differentiation (Diff) on days 1 and 14. a** Light microscopic images; **b** 10 μm cryosections of static spheroids with cell death markers for apoptosis (cCas3), necrosis (RIPK3), and hypoxia (Hif-1α);

**c** 10 μm cryosections of static spheroids with PPARγ; **d** 10 μm cryosections of static spheroids with perilipin 1 expression; **e** 10 μm cryosections of static spheroids with accumulated lipids; cell death markers: orange, PPARγ: green, perilipin 1: green, lipids: yellow, cell nuclei: white; scale bar: 200 μm.; $n = 3$ biological replicates.

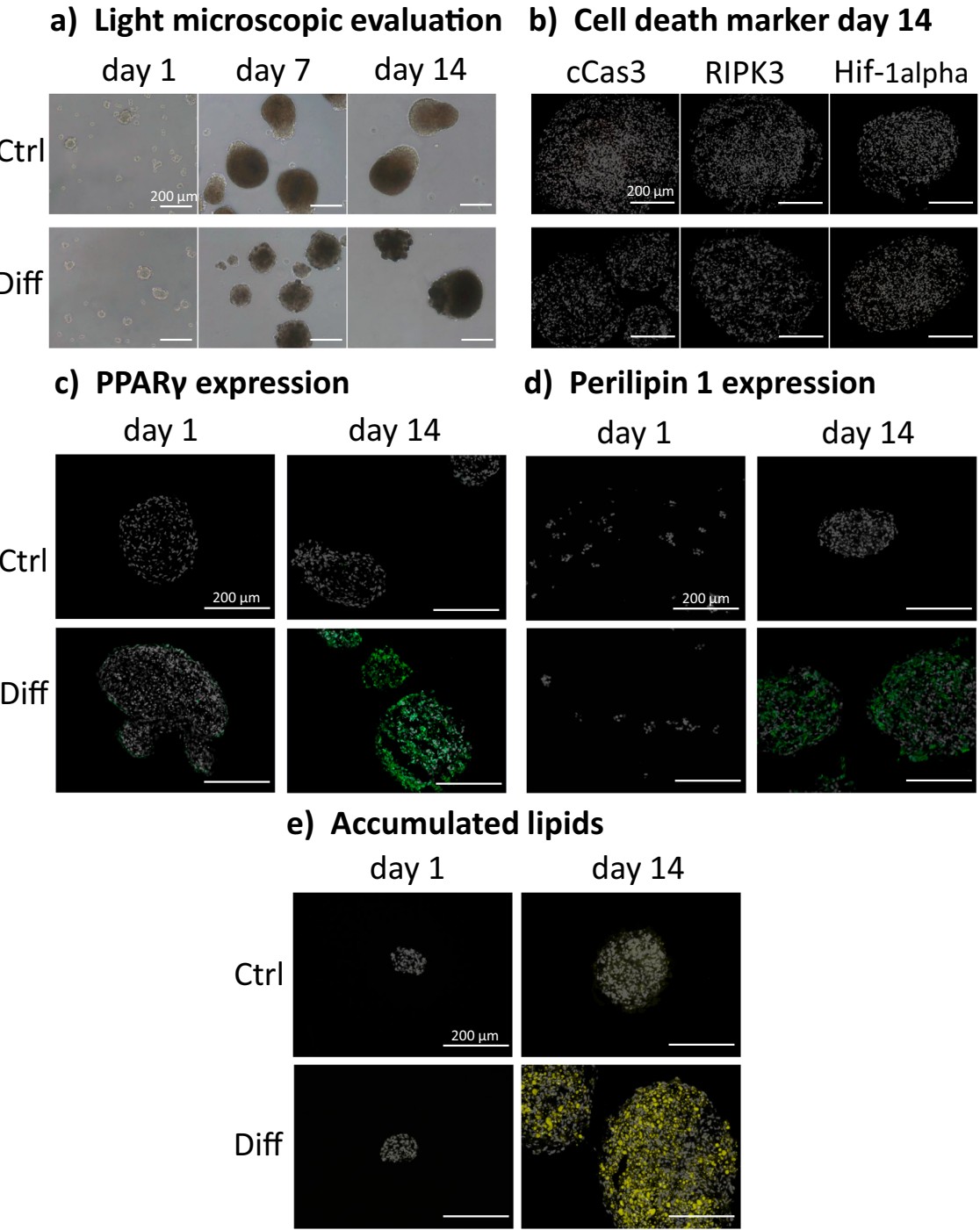

**Fig. 4 | Evaluation of dynamic bASC spheroid culture during maintenance (Ctrl) and adipogenic differentiation (Diff) on days 0 and 14. a** Light microscopic images; **b** 10 µm cryosections of dynamic spheroids with cell death markers for apoptosis (cCas3), necrosis (RIPK3), and hypoxia (Hif-1α).; **c** 10 µm cryosections of dynamic spheroids with PPARγ; **d** 10 µm cryosections of dynamic spheroids with perilipin 1 expression; **e** 10 µm cryosections of dynamic spheroids with accumulated lipids; cell death markers: orange, PPARγ: green, perilipin 1: green, lipids: yellow, cell nuclei: white; scale bar: 200 µm.; *n* = 3 biological replicates.

## Fatty acid composition of bASC spheroids and bovine fat

In Fig. 6a, macroscopic images of native bovine fat tissue and our in vitro produced and differentiated fat spheroid biomass after harvest are provided. Looking at the nutritional value of the bASC spheroids, significant differences in the fatty acid composition between native bovine fat tissue, differentiated spheroids and control spheroids were revealed. While the fat in all samples mostly consisted of saturated fatty acids (SFA), the proportion of saturated, monounsaturated and polyunsaturated fatty acids (MUFA and PUFA) differed considerably (Fig. 6b and Supplementary Table 2). Native bovine fat was found to consist of 82.2% SFA. Control spheroids came close to the native fat with 83.9% SFA in their fat content. The fat in differentiated spheroids had significantly less SFA content, only 66.2% SFA (Fig. 6c). MUFA made up about 16.5% of fatty acids in bovine fat but only 5.8% in the fat content of the control spheroids. In differentiated spheroids, 27.3% of the fat content were MUFA, significantly more than in native fat and control spheroids. PUFA are only represented with 1.2% in bovine fat. In control spheroids about 9.7% and in differentiated spheroids 5.8% PUFA were found.

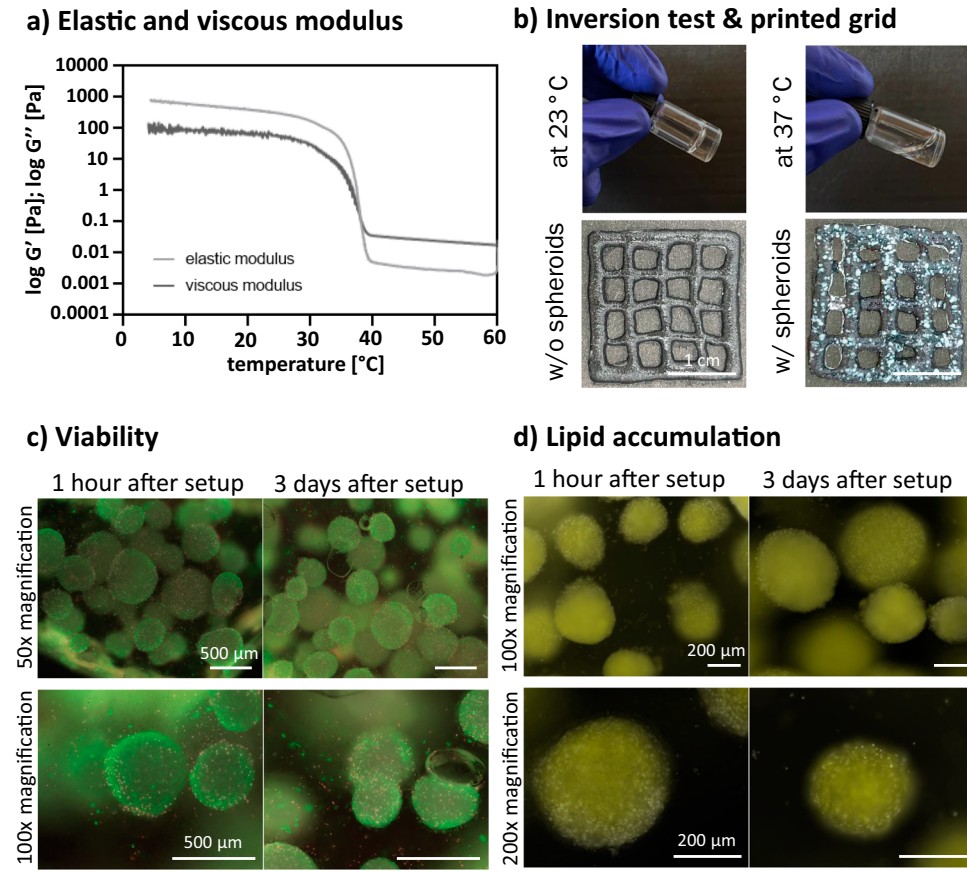

**Fig. 5 | 3D bioprinting of differentiated bASC spheroids in GG-based bioink.**
**a** Elastic and viscous modulus of 1.5 % GG at 37 °C.; **b** Inversion test of 1.5 % GG at 25 °C and 37 °C and printed macroscopic grid (20 × 20 × 2 mm) without (w/o) and with (w/) differentiated bASC spheroids. Scale bar: 1 cm; **c** Viability of bioprinted bASC spheroids after 1 h and 3 days after setup.; viable cells: green; dead cells: red; cell nuclei: white, scale bar: 500 μm; **d** Lipid accumulation in printed bASC spheroids 1 h and 3 days after setup.; lipids: yellow; cell nuclei: white, scale bar: 200 μm.; n = 3 biological replicates. Source data are provided as a Source Data file.

Having a closer look at the fatty acid spectrum, SFA were quite abundant in all samples with palmitic acid (C16:0) as the most common fatty acid in all samples ranging from 49.7% in bovine fat to 41.7% in control spheroids and 41.5% in differentiated spheroids (Fig. 6d). While stearic acid (C18:0) was the second most present fatty acid in bovine fat tissue and control spheroids, with 19.2% and 38.2% respectively, it ranked in place three in differentiated spheroids with 21.6%. A relevant quantity (9.0%) of myristic acid (C14:0) was only present in bovine fat. Other SFA were present in the samples as well but were only found in small amounts (Fig. 6d). By far the most common MUFA in all samples was oleic acid (C18:1 cis), with 24.5% in differentiated spheroids, but way less in bovine fat (12.7%) and control spheroids (5.3%). The only other MUFA that was found in a small but non-negligible amount in bovine fat and differentiated spheroids was palmitoleic acid (C16:1) with 2.5% and 2.4% respectively. PUFA was the least abundant fatty acids in all samples. The largest amounts were found in the spheroid samples and were identified as arachidonic acid (C20:4 delta 5,8,11,14) and/or dihomo-γ-linolenic acid (C20:3) with 7.2% in control spheroids and 4.4% in differentiated spheroids (Fig. 6e).

## Discussion

To pursue the goal of in vitro-grown near-natural bovine adipose tissue, we decided to develop a versatile building block suitable for many different applications. We utilized primary cells derived from adult Limousin cows, freshly isolated post-slaughter. While primary cells offer certain advantages as a cell source, such as their natural characteristics, they also present limitations, such as donor variability. Looking ahead, the identification of suitable cell sources, particularly pluripotent stem cells, will be crucial for future mass production and potential market entry of CM[38]. Despite the challenges in handling and cultivation associated with a primary cell source, we have been able to demonstrate our proof of concept using our developed differentiation medium for adipose-derived stem cells to produce fat spheroids. Furthermore, we demonstrated that these spheroids can serve as building blocks for the biofabrication of CM constructs using 3D bioprinting techniques. Alternatively, their use via bioassembly is also conceivable, for example, using methods such as the kenzan technique[39]. The following discussion outlines the individual steps involved in the production and characterization of differentiated bASC spheroids, as well as their utilization as building blocks, using extrusion-based bioprinting.

First, a simple one-step differentiation protocol for bASCs in 2D and 3D culture was developed. While indomethacin was able to induce weak adipogenic differentiation in bASCs, it was less effective than the insulin sensitizer rosiglitazone. The elimination of IBMX in the differentiation mix led to no visible changes in lipid accumulation. These results correspond to recent findings of Mitić et al. who revealed the redundancy of dexamethasone and IBMX in bovine adipogenesis, as both are activating the transcription of PPARγ via a similar pathway[40]. With the remaining three differentiation factors, rosiglitazone, dexamethasone and bovine insulin, we were able to show successful differentiation of bASCs in a single step. In both 2D and 3D cell culture, cells showed characteristic lipid filled vacuoles.

Also, eliminating IBMX in the differentiation mix helps to reduce production costs and the use of potentially harmful substances. Still, there is further need for a more sustainable and FBS-free CM

## a) Macroscopic images of native fat tissue and differentiated spheroids

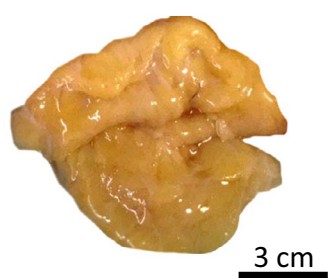

3 cm

**Native bovine fat tissue**

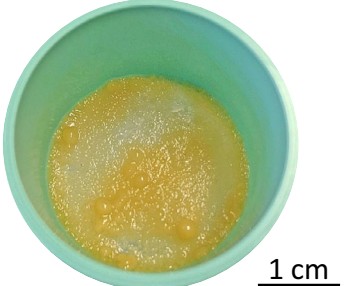

1 cm

**Differentiated fat spheroids**

### b) Fatty acid composition

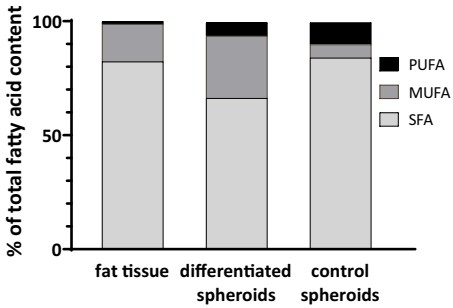

### c) Fatty acid composition in detail

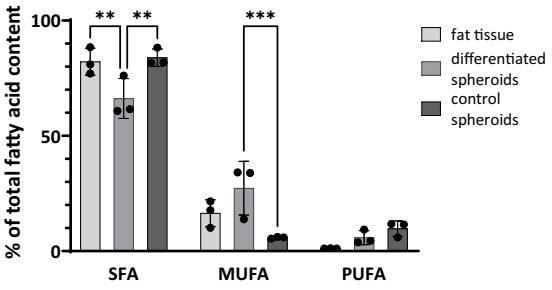

### d) Fatty acid spectrum

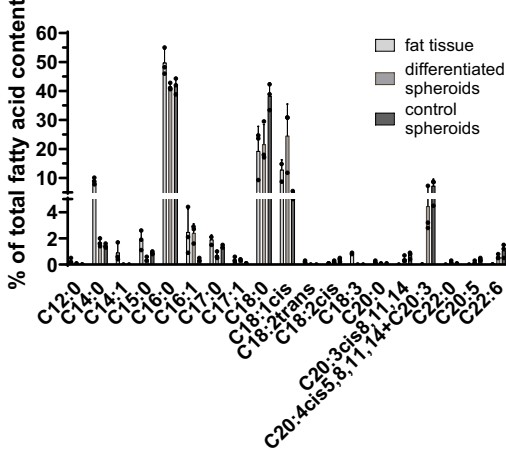

### e) Most abundant fatty acids

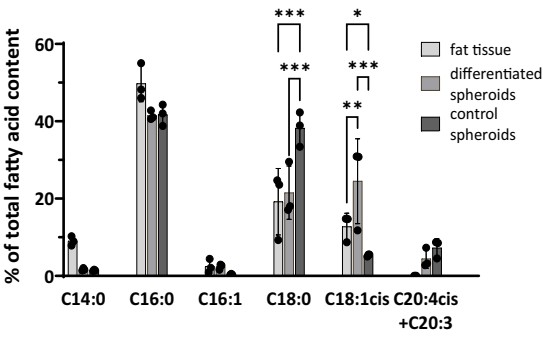

**Fig. 6 | Fatty acid composition of spheroids and native bovine fat tissue.**
**a** Macroscopic images of native bovine fat tissue (left) and solely dynamically cultured and differentiated fat spheroids after harvest (right). Scale bar: 3 cm and 1 cm; **b** Overview of saturated (SFA), monounsaturated (MUFA) and polyunsaturated (PUFA) fatty acids in differentiated (Diff) and control (Ctrl) spheroids and bovine fat tissue (Fat).; **c** Fatty acid composition in detail. Significances are noteable for SFA for Fat vs. Diff ($p = 0.0077$) and Diff vs. Ctrl ($p = 0.0052$). MUFA: Diff vs. Ctrl ($p = 0.0009$); **d** Fatty acid spectrum of all samples.; **e** Most abundant fatty acids. Significances are noteable for C18:0 for Fat vs. Ctrl ($p < 0.0001$) and Diff vs. Ctrl ($p < 0.0001$). C18:1cis: Fat vs. Diff ($p = 0.0017$), Fat vs. Ctrl ($p = 0.0393$), Diff vs. Ctrl ($p < 0.0001$).; All data are plotted as mean and for (**c**–**e**) ± SD. Dots represent single data points.; Significances were calculated using two-way analysis of variance and Holm-Šídák multiple comparisons test; *: $p \leq 0.05$, **: $p \leq 0.01$, ***$p \leq 0.001$; $n = 3$ biological replicates. Source data are provided as a Source Data file.

production[40,41]. In the future, artificial differentiation factors could be substituted by compositions of edible and affordable components such as fatty acids as shown by Kang et al. and Mehta et al.[42,43].

Second, we established a static and dynamic spheroid culture of primary bASCs. In static culture, spheroids were stable for at least two weeks. The size of control spheroids decreased over culture time as seen in other MSC spheroids[44,45]. Differentiated spheroids were larger

compared to the control, due to evenly spread lipid accumulation within the spheroids. Combined with PPARγ, perilipin 1, and lipid staining and subsequent quantification, the differentiation and the presence of lipid-filled vacuoles were proven. Since cell death was not detected within 14 days of static or dynamic culture, the spheroids were found vital and stable. The generation of spheroids from bovine adipogenic precursors in static culture is scarcely published and the

differentiation within the spheroids is even less examined. While Wang et al. were able to generate static spheroids via hanging drop culture, they only differentiated the cells after the spheroids adhered to the surface of a cell culture plate[46]. Naraoka et al. differentiated spheroids from a CD29 positive subpopulation of isolated MSCs from bovine muscle tissue. Those spheroids were able to undergo adipogenic differentiation in a static system[47]. Similar to our results, the spheroids showed lipid vacuoles in different sizes.

In general, dynamic spheroid culture of MSCs, human or animal sourced, is a complicated process and not too many cell types are published at all to be cultured that way[48,49]. In this work, we differentiated bovine fat spheroids in dynamic culture. Our results could lead to scaffold-free, scalable and dynamic culture approaches in bioreactors for CM production. We show that bASCs can form spheroids in dynamic culture and retain their ability to differentiate and form lipid vacuoles. In general, spheroids offer many important advantages compared to 2D cultures. The 3D structure offers a more in vivo like microenvironment, 3D cell-cell contacts and enables a more realistic metabolic activity of the cells. Apart from the outer cell layer the cells within a spheroid are also partly protected from mechanical influences. These properties make spheroids the perfect building block for tissue engineering[36,50,51].

Biofabrication, with its automated techniques, offers a significant opportunity to process cell mass into CM products in the future[39]. Hence, we tested the bioprintability of spheroids in edible GG after 14 days of differentiation in dynamic culture. GG is an exopolysaccharide produced by fermentation of, for example, corn starch by bacteria from the *Sphingomonas* group[52]. Therefore, it can be produced without animal ingredients and is already approved as a food additive, mainly as gelling agent with very low allergy risk[53,54]. GG also can be modified by adding plant-based proteins to improve its nutritional value[55]. Furthermore, GG demonstrated favorable characteristics for 3D culture, bioprinting, and differentiation of primary human ASCs[56]. All these properties make GG a promising candidate as an ingredient for CM. Bioprinting of bovine fat spheroids has not been done before, but Colle et al. printed differentiated human ASC spheroids in gelatin methacryloyl[57]. Our spheroid viability results are in accordance with the findings of Colle et al., who also found satisfying cell viability and spheroid stability after the 3D bioprinting process and no influence on the lipid distribution within the spheroids. Looking at our results, we are positive that we present data highlighting bASC spheroids as a very interesting and promising building block for sizable bioprinted CM products.

The analysis of the fatty acid composition of the bASC spheroids gave a more in-depth picture of their potential in cultured fat engineering. It is known that the fatty acid composition in cells in vitro is dependent on the culture medium and consequently differs from cells in vivo[58]. Therefore, differences between spheroids and native bovine fat tissue were to be expected. Moreover, disparities between differentiated spheroids and the undifferentiated spheroid control were found. While control spheroids show roughly the same amount of SFA as found in native fat, differentiated spheroids display a significantly reduced amount of SFA compared to the other groups.

The significantly elevated amount of MUFA in differentiated spheroids is mostly caused by an increased level of oleic acid (18:1cis). In accordance with the literature, oleic acid is the most common MUFA in the native bovine fat tissue samples[59,60]. This is also true for all spheroid samples, seeing the highest percentage in differentiated spheroids and significantly lower amounts in control samples. The lower amount of SFA in differentiated spheroids is caused by decreased amounts of C14:0, C16:0 and C18:0. Control spheroids showed a similar decrease in C14:0 and C16:0 but a significant increase in C18:0 compared to native fat. In summary, spheroids presented a fatty acid profile significantly different from native fat tissue. The differentiated spheroids had a significantly higher content of C18:1cis and slightly lower contents of C16:0 and C14:0, making the fatty acid composition potentially healthier compared to native fat[61]. In future experiments the fatty acid composition could be tuned to obtain a healthier or a more natural spectrum, opening further opportunities for CM production.

Taking all our results into consideration, we are positive that primary bASC spheroids are highly suitable as versatile building blocks for cultured fat production. They provide the advantage of a more in vivo-like 3D structure and a high cell density, making them particularly suitable for additive manufacturing of CM products. Additionally, they can be cultured and differentiated without the use of antibiotics, in static and dynamic culture systems and offer the potential for tailoring the fatty acid composition in future applications.

## Methods

### Tissue source

For the experiments, fat tissue was sourced from 2-year-old freshly slaughtered cattle (Limousin) from a local organic butcher shop. The fresh tissue was transported for 30–40 min in PBS+ (PBS-2A with calcium and magnesium, Capricorn Scientific GmbH, Ebsdorfergrund, Germany) with 0.2% Primocin (Ant-pm-2, InvivoGen, San Diego, CA, USA) at room temperature (RT).

### Cell isolation bASCs

Fat tissue was transferred to a glass plate. Before cutting the tissue into 2-5 mm sized pieces, remainings of different tissue types were removed using a ceramic knife and scissors. Then, the chopped tissue was mixed with the same volume (w/v) of a collagenase type I solution (520 U/mL; 10114532, Fisher Scientific GmbH, Schwerte, Germany) in DMEM high glucose (DMEM hg: L0101-500, VWR International GmbH, Radnor, PA, USA) with 0.2% Primocin and digested for 3 h at 37 °C on a shaker (~150 rpm). Next, the digested tissue was separated from undigested tissue and the stromal vascular fraction by centrifugation for 10 min at 700 g. The supernatant was discarded, and the pellet was resuspended in 10 mL erythrocyte lysis buffer (155 mM $NH_4Cl$, 21235.297, VWR international GmbH; 10 mM $KHCO_3$, 1.04854.0500, Merck KGaA, Darmstadt, Germany; 0.1 mM EDTA, 15575-038, Invitrogen, Waltham, MA, USA) before adding the same volume of DMEM hg after 5 min of incubation. This suspension was centrifuged for 5 min at 700 g. In the following step, the supernatant was removed, and the pellet was resuspended with 10 mL PBS +. Afterwards, the cell suspension was strained using a 70 μm cell strainer (732-2758, VWR International GmbH). The vessel and the cell strainer were washed with 10 mL PBS +. The filtered suspension was centrifuged for 10 min at 700 g. The supernatant was discarded, and the cell pellet was resuspended in media for cell counting. The cells were seeded at a density of 20,000–40,000 cells/cm² in a cell culture-treated T-flask (Greiner Bio-One GmbH, Frickenhausen, Germany) in DMEM hg complemented with 2 mM glutamine (882027, Biozym Scientific GmbH, Hessisch Oldendorf, Germany) and with 10% fetal bovine serum (FBS, FBS Gold South American origin, PAN-Biotech GmbH, Aidenbach, Germany). After 1 day, the cells were washed, and new medium was added.

### Culture conditions

The isolated bASCs were cultured in DMEM-FBS. 0.2% Primocin was added to the medium until the first passage. After that, the cells grew in an antibiotic-free medium. Twice a week, the medium was changed. For passaging, 0.05% trypsin (11570626, Fisher Scientific GmbH) dissolved in 0.53 mM EDTA solution was used after cells reached about 90% confluence.

### Growth kinetics

For the bASCs, growth kinetics were determined. 10,000 cells/well were seeded in each well of a 24-Well plate (662160, Greiner Bio-One GmbH). For 10 days, each day, the cell count was determined by cell

counting of triplicates of each of the three different donor animals. For the Doubling time (DT) firstly, the frequency of cell cycles per day (f; [1/ d]) was calculated with the cell number at the end of exponential growth ($N_t$), the cell number at the start of the exponential growth ($N_0$) and the time of exponential growth (t, [d]). Next, f was converted to DT. The following equations were used for the calculation:

$$f = \frac{\log(\frac{N_t}{N_0})}{\log(2)*t} \qquad (1)$$

$$DT = \frac{24}{f} \qquad (2)$$

### Final adipogenic differentiation

Adipogenic differentiation was induced through the supplementation of rosiglitazone (2.5 µM, Caym71740, Cayman Chemical), bovine insulin (3 µg/mL, I0516), and dexamethasone (1 µM, D4902, both Sigma-Aldrich, St. Louis, MO, USA) to DMEM (DMEM-Diff). A complete media change was done thrice per week.

### Static spheroid culture

For static spheroid formation, 150 µL anti-adherence rinsing solution (STEMCELL Technologies, Vancouver, Canada) per well of a 96 well u-bottom plate (650185 Greiner Bio-One GmbH) was incubated for 2 min. Afterwards, wells were washed with 200 µL PBS+ each. Then 100 µL of cell suspension with a defined number of bASCs (10,000; 25,000; 50,000) in DMEM was pipetted into each well, with the outermost well rows filled with PBS + , to prevent evaporation. Spheroid formation took place within 2–3 days at 37 °C and 5% $CO_2$ atmosphere. Spheroids were cultured in DMEM or DMEM-Diff with a change of 70% of the media volume three times a week for 14 days.

### Dynamic spheroid culture

Cell culture dishes (10 mm CytoOne, untreated, StarLab International GmbH, Hamburg, Germany) were treated with 15 mL anti-adherence rinsing solution and incubated for 2 min on an orbital shaker at 90 rpm (OHAUS SHLD0415DG, VWR International GmbH). Afterwards, dishes were washed with 15 mL PBS + . bASCs were seeded with a cell density of 250,000 cells per 1 mL DMEM. For spheroid formation, cells were placed on an orbital shaker at 70 rpm for two to three days. Afterwards, culture medium was changed to DMEM-Diff or cells were further cultured in DMEM. Refreshing the medium thrice a week was done by collecting the cellular suspension and centrifuging it at 500 g for 2 min. The supernatant was discarded, and fresh media was used to resuspend the spheroids. Finally, the whole volume was given in the previously used cell culture plate and placed on the orbital shaker with 80 rpm at 37 °C and 5 % $CO_2$.

### Fixation and cryosectioning

Prior to fixation, cell culture medium was discarded, and samples were washed with PBS + . Cells in 2D were fixed for 15 min using Histofix (P087.5, Carl Roth GmbH + Co.KG, Karlsruhe, Germany). Spheroids were fixed for 30 min after incubation in 30% sucrose solution for 30 min, and bioprinted spheroids were fixed for 60 min in Histofix. 2D and 3D samples were washed twice with PBS+ after fixation. For cryosectioning, spheroids were embedded in mounting medium (6502B; Thermo Fisher Scientific, Waltham, USA) within small molds (7 × 7 mm; 27147-1, Ted Pella Inc., Redding, USA) on dry ice. Afterwards, samples were cooled down to −80 °C and sliced into 10 µm thick sections at a chamber temperature of -35 °C on a cryotome (CM3050S, Leica Biosystems Nussloch GmbH, Germany).

### Immunofluorescence

Fixed cells in well plates were permeabilized for 5 min with 0.1% Triton X-100 (T8787, Sigma-Aldrich, St. Louis, MO, USA) in PBS+ and washed once with PBS+ . Permeabilization was prolonged to 30 min for bioprinted constructs, and the washing step was extended to 15 min. For previously cryosectioned spheroids, this step was omitted. Afterwards, samples were blocked with 1% BSA (01400.100, Biomol GmbH, Hamburg, Germany) in PBS+ for 30 min at RT, followed by incubating the primary antibody at 4 °C overnight. The cells were washed three times with PBS+ for 5 min and incubated with the secondary antibody for 45 min at RT. Before cell nuclei staining with DAPI (1 µg/mL;18860.01 SERVA Electrophoresis GmbH, Heidelberg, Germany) for 10 min, cells were washed three times with PBS+ for 10 min. All images were acquired by a fluorescence microscope (Axio Observer) with an Axiocam 305 using ZENblue (all Carl Zeiss Carl Zeiss AG, Jena, Germany).

Primary antibodies: CD56 (1:50; 304602, BioLegend, San Diego, CA, USA), CD73 (1:50; 12231-1-AP, Proteintech Germany GmbH, Planegg-Martinsried, Germany), CD90 (1:50; 66766-1-Ig, Proteintech Germany GmbH), CD105 (1:50; 10862-1-AP, Proteintech Germany GmbH), perilipin 1 (1:200; Ab 3526 Abcam, Cambridge, UK), PPAR Gamma (1:200; 16643-1-AP, Proteintech Germany GmbH), cCas3 (1:400, 9661 Cell Signaling), RIPK3 (1:200, LS-C354158, LifeSpan BioSciences Inc.), and Hif-1α (1:50; 14-9100-80, Invitrogen).

Secondary antibodies: anti-mouse (1:200; Ab150113) or anti-rabbit AlexaFluor 488(1:200, Ab150077, both Abcam, Cambridge, UK), anti-mouse Cy3TM3 IgG (1:200, 115165003, Dianova GmbH, Hamburg, Germany), and IgG control antibodies for monoclonal mouse IgG1 (Sc-3877) and IgG2a (Sc-3878, both Santa Cruz Biotechnology) and polyclonal rabbit IgG (NI01, Merck KGaA).

### Live-dead staining

For live-dead staining, the samples were incubated in a dye solution containing 2 µM calcein, 4 µM ethidium homodimer-1 (10237012, Fisher Scientific, Schwerte, Germany), and 1 µg/mL Hoechst 33342 (40825, Cell Signaling Technology Europe B.V., Frankfurt, Germany) in PBS+ for 60 min at RT. Fluorescence microscopy followed immediately.

### BODIPY staining

The fixed cells and bioprinted constructs were washed three times with PBS+ and then stained at RT in the dark with BODIPY 493/503 (1 µg/mL, Cay25892-10, Cayman Chemicals, Ann Arbor, USA) and Hoechst 33342 (1 µg/mL) diluted in PBS + . After 1 h, the cells were washed three times with PBS + . Fluorescence microscopy followed immediately. The quantification of the BODIPY/Hoechst staining of differentiated adherent cells in 2D culture was carried out by analyzing fluorescence images via MATLAB (version R2022a, MathWorks, Natick, USA). A custom script was used to calculate the average fluorescence signal (BODIPY or lipids) and the number of cell nuclei (Hoechst). To compare different conditions, a ratio of the green signal to cell nuclei was determined to obtain the average green value per cell.

### Actin staining

Before the ActinRed 555 ReadyProbe (15119325, Fisher Scientific, Schwerte, Germany) staining, cells were permeabilized for 5 min with 0.1% Triton X-100 in PBS + , washed once with PBS+ and incubated in a staining solution with 1 droplet ActinRed and 1 µL Hoechst 3342 (1 µg/ mL) for 45 min. Afterwards, samples were washed two times with PBS + . Fluorescence microscopy followed immediately.

### Bioprinting and model culture

For bioprinting, adipogenic differentiated spheroids were encapsulated into a 1.5% (w/v) gellan gum (Gelzan™, CP Kelzo G1910, Sigma-Aldrich) bioink and printed with a BIO X with a temperature-controllable printhead (Cellink AB, Gothenburg, Sweden). Therefore,

spheroids from dynamic culture, started with a cell number of $36 \times 10^6$ cells, were collected and centrifuged for 2 min at 500 g. The spheroids were suspended in 300 μL PBS + . This suspension was homogeneously mixed into 2.7 mL previously dissolved GG solution (150 mg boiled in 9 mL $H_2O$) at a temperature of 37 °C to obtain a bioink with a cell concentration of $12 \times 10^6$ cells per 1 mL. The GG bioink was transferred into a 3 mL cartridge (Cellink AB) and placed into the printhead with a pre-set temperature of 25 °C. Bioprinting was performed with an applied pressure of 10 kPa at a temperature of 25 °C and a velocity of 20 mm/s. The resulting model was a $20 \times 20 \times 2$ mm grid in a 6-well plate (Cellstar TC, Greiner Bio-One GmbH). After bioprinting, cross-linking with cell culture medium for 60 min followed. The models were analyzed immediately after printing and after a culture period of three days in differentiation medium. The microscopy images were taken using the z-stack method.

### Spheroid size evaluation

Phase contrast images of the spheroids were analyzed concerning the size of the spheroids using a macro written with ImageJ FIJI 1.53c Java 1.8.0_172 and supported by Read_And_Write_Excel (GitHub, version 1.1.6). First, the data were converted into an 8-bit format to set the threshold, so spheroids appear black against the white background. Possible irregularities of the spheroid, which could be perceived as white holes, were corrected by the function, fill holes. Afterwards, the pre-set parameters, area, and diameter were calculated and exported as an Excel file and tagged image file. The coordinates of the analyzed object were saved as.roi (region of interest) file, and the individual values as.csv (comma-separated values) file.

### Quantification of spheroid differentiation

The fluorescence intensities for the staining of PPARγ, perilipin 1, and BODIPY were assessed using the ZENblue software (Version 2.3). In this process, fluorescence images of static or dynamic spheroid cryosections were utilized and a frame was defined around each spheroid. The mean intensity of the fluorescence signal within each frame was recorded. Subsequently, all values were normalized to the mean value of the corresponding day 14 controls.

### Determination of fatty acid composition

Fatty acids were measured as fatty acid methyl esters (FAME) via GC-MS/MS. Transesterification was carried out according to a modified protocol[62–64] and is described as follows. Spheroid samples were homogenized at a concentration of 100 mg/mL in 0.9% (w/v) NaCl solution. Aliquots of 200 μL homogenized sample were transferred into tubes, and the aqueous phase was evaporated overnight in a fume hood. 2 mL of 1% (v/v) sulfuric methanol was added. The samples were vortexed and subsequently heated to 80 °C for 1 h. During incubation, samples were sonicated 3 times for 5 min. After incubation, the samples were put on ice and 0.5 mL of saturated NaCl solution and 0.5 mL $H_2O$ were added. For FAME extraction, 2 mL hexane was added, and the samples were mixed by shaking vigorously. After phase separation, 500 μL of the hexane supernatants were transferred to brown glass vials. Prior to GC-MS analysis, 10 μL of 0.2 mg/mL ethyl-myristate was added as internal standard. All samples were derivatized and analyzed in duplicate.

GC/EI-MS/MS analysis was carried out on an Agilent 7890B gas chromatograph coupled with an Agilent 7000D triple quadrupole mass spectrometer (Agilent, Waldbronn, Germany). The injection volume was 1 μL (splitless). The separation was done on a Rtx-2330 fused silica capillary column (60 m, 0.25 mm I.D., 0.1 mm film thickness; Restek, Bad Homburg, Germany) and the injector temperature was set to 250 °C, carrier gas was He with a flow rate of 1.2 mL min⁻¹. The temperature program was 60 °C (1 min), followed by an increase of 6 °C min⁻¹ to 150 °C, 3 °C to 190 °C and 6 °C to 250 °C (5 min). The transfer line and source temperature were set to 230 °C. The mass selective detector was operated in SIM mode with the following masses: m/z 101, 88, 87, 81, 79, and 74.

FAME signals were assigned via a 37 component FAME standard (Supelco, Bellefonte, PA, USA) and FAME composition was expressed as percentage of total FAME.

### Statistics

All data are from three independent experiments. The shown values are expressed as mean $\pm 1^{st}$ standard deviation (SD) if not stated otherwise. Testing of normality distribution was performed using the Shapiro-Wilk test. Significances were determined by two-way analysis of variance (ANOVA) followed by Holm-Šídák multiple comparisons test or Welchs t-test in GraphPad Prism 10.0.0. Results were considered statistically significant with $*P \le 0.05$; $**P \le 0.01$; $***P \le 0.001$.

### Reporting summary

Further information on research design is available in the Nature Portfolio Reporting Summary linked to this article.

## Data availability

The data generated in this study are provided in the Supplementary Information/Source Data file. Source data are provided with this paper. That includes the data of the findings in Fig. 2a (cell yields) and 2b (growth kinetic), Fig. 5a (elastic and viscous modulus), Figs. 6b–e (fatty acids analysis), Supplementary Figs. 1a-d, 2b-d and 3a. Source data are provided in this paper.

## Code availability

The code for the analysis of our BODIPY fluorescence images via MATLAB is provided here: https://github.com/harkonsen/BODIPY.

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

## Acknowledgements

First and foremost, we thank the Avina Foundation and New Harvest for their generous support, which made this research possible. The Avina Foundation (Hurden, Switzerland) has financially supported all authors of this work equally within the project "Fleischsphäroide— Grundbausteine zur nachhaltigen Herstellung von gesunden in vitro Fleischprodukten". New Harvest supported the work of J.O.W. with a scholarship. Further, we want to thank the local organic butcher shop Griesshaber, especially Desirée Griesshaber-Vetter for providing us with bovine fat tissue. We thank Antje Eschle, Jonas Pospiech, and Antonia Heuser for their outstanding technical support. Finally, our thanks go to Prof. Blaeser and Robin Maatz from the University of Darmstadt for their kind help and Prof. Jan Frank and Dr. Monika Bach for the vivid discussions.

## Author contributions

P.J.K., A.K., and S.H. conceptualized and designed the study; A.K., J.O.W., F.B.A., T.M., H.H., S.R., and L.B.H. performed experiments; A.K., J.O.W., F.B.A., T.M., H.H., S.R., and L. B. H. analyzed the data; P.J.K., A.K., J.O.W., and S.H. supervised data analysis and visualization; A.K., J.O.W., F.B.A., and S.H. prepared figures, A. K., J.O.W., F.B.A., T.M., and H.H. performed statistical analysis; A. K., J.O.W., F.B.A., H.H., S.H., and P.J.K. wrote the initial draft of the manuscript, A.K., J.O.W., F.B.A., H.H., S.H., and P.J.K. revised the manuscript.

## Funding

## Competing interests

The authors declare no competing interests.
