## [Peer Review File · Nature Communications]

Reviewers' Comments:

Reviewer #1:

Remarks to the Author:

Major comments:

The growth and adipogenic differentiation of bovine adipose-derived stem cells in spheroids under dynamic conditions is an emerging topic for academic and industry players in the cultivated meat field.. The authors have done a good job of performing experiments ranging from cell proliferation, adipogenic differentiation, spheroid culture, dynamic culture, 3D printing, and nutritional profiling. However, many of the results have been redundant from previous publications; for example, spheroid culture (including bovine ASCs; many using human ASCs), reduced adipogenic cocktail, fatty acid composition, and 3D printing/construction of adipocytes. It is thus questioned that the manuscript would significantly advance the technological aspect of cultured fat research. The reviewer feels that it fits better for more specialized journal. Some of the comments on the contents are described below; there are many 'minor' comments for their future reference as the writing needs further substantial improvement.

1. Many parts of the article are not clearly explained. It is very hard for the reader to understand.
2. Apart from showing Bodipy and perilipin 1 expression, authors can also supplement with the expression of other adipogenic markers such as PPAR γ and FAS. It can be intracellular staining, western blot or qPCR.
3. The author state that rosiglitazone showed higher bodipy staining as compared to indomethacin (Figure 2D). However, there is no difference in fluorescence intensity per cell (Supplementary Figure 1B). Is there any issue with the image analysis or a result of huge variation? The data from Supp Figure 1B does not look so convincing.
4. The authors claim the dynamic culture is novel, but I am not sure if the experiment is convincing enough. They should have sufficient data to show why it works better than the static culture. Any quantifiable data to show the trend over time?
5. If the reduction in size of bASC spheroid is not caused by cell death, what caused it?
6. For nutritional profiling, what is the absolute concentration of the major fatty acids? The cells may have similar composition (e.g. 60-80% SFA), but differ in absolute amount (e.g. 0.2g of SFA per g of cells). This is very important as it has implication on feasibility of this entire technology.

Minor comments

Line 1: encourage the author to use "cultivated meat" instead of "cultured meat" as an industry norm.

Line 30: Suggest to remove "without the need to raise the whole animal". It might be

benefit of cultivated meat. However, it does not add any value towards the main message of growing cultivated meat research field.

Line 33: In future, it could contribute to the nutrition of the [remove the] growing world population while producing less [lesser] greenhouse [gas] emissions and using less [lesser] land than conventional animal husbandry.

Line 38: what are the “hurdles in all parts of the process”?

Line 45: “ Usually, adherent cells ... making it time, work, and material consuming” This sentence is too long and contains too many idea. Please revise it.

Line 48: “When employing scaffolds such as non-edible microcarriers, an additional step is necessary to harvest the cells before proceeding with further processing”.

Change to “When employing non-edible scaffolds such as microcarriers, an additional step is necessary to harvest the cells before proceeding with further processing”. There are scaffolds that are edible and do not require harvesting or further processing. Thus it is important to be specific to non-edible scaffold.

Line 52: What is “so-called spheroids”? If they are spheroids, just term the as spheroid. If they are not, please refrain from using the term to confuse readers.

Line 58: “However, very little research has been conducted concerning bovine fat spheroids”. Do not have sufficient link to the previous sentences. It might be meaningful to explain the importance of fat spheroid culture or research done on spheroid from other animal or tissue origin?

Line 65: “The development of a new dynamic spheroid culture of bASCs is a delicate process” to change to “The development of a new dynamic spheroid culture of bASCs for cultivated meat is a delicate process”. Authors need to highlight cultivated meat here as such spheroid culture can be used for other purposes that do not need to address challenges of cultivated meat research and challenges.

Figure 1: font used in the figure looks very different. I would suggest to change to font that are easier to read.

Figure 1 caption: the following sentences are incomplete. Please rephrase them “Technological, financial, consumer, and regulatory gaps that need to be addressed for cultured meat commercialization” and “Following, characterization, bioprinting and fatty acid analysis.”

Line 84: change “varied within the 84 subcutaneous fat for the” to “varied between the 84 subcutaneous fat from the”

Line 92 – 104: it is very difficult to understand or follow the author. I will suggest to rephrase them. Also, it appears that indomethacin and IBMX might not be needed for adipogenic differentiation. However, did the author investigate if dexamethasone and insulin are needed?

Line 113: What caused the difference in spheroid size after 5 days? The author showed that the expression of cCas3, RIPK3 and Hif- α were absent on Day 14. Did the author investigate look at the expression of cCas3, RIPK3 and Hif- α at different timepoint?

What is the message or conclusion that the author is trying to show?

Line 141: “The size distribution was most uniform when maintaining a culture density of 250,000 cells/mL at a shaking speed of 70 rpm during spheroid formation” How did the author measure the size distribution?

Line 143: “For spheroid maintenance and differentiation, a shaking speed of 80 rpm yielded the most favorable results” What did the author mean by most favorable results? Size distribution, cell yield, differentiation efficiency, lipid uptake? What is the range of shaking speed that the author investigated?

Line 147: $3.68 \text{ mg} \pm 1.9 \text{ mg}$ has a huge deviation. Is there any way to reduce the deviation?

Figure 4A: In Day 14, control and differentiated spheroids look similar in size. Why is it that differentiated spheroids look 2x larger than control spheroids in Figure 4C and 4D? Figure 5C and 5D: the images look out of focus. Will z-step or confocal imaging be meaningful?

Line 195: “While all samples contained mostly saturated fatty acids (SFA) the proportion of saturated, monounsaturated and polyunsaturated differs considerably (Figure 6 B).”
to change to “While all samples contained mostly saturated fatty acids (SFA), the proportion of saturated, monounsaturated and polyunsaturated fatty acids differs considerably (Figure 6 B).”

Line 196 – 202: it is very difficult to follow. Please kindly rephrase to help the readers to understand easier.

Figure 6B and 6C: both figures are showing fatty acid composition with different representations. Please amend the figure caption for 6C.

Figure 6D and 6E: same as above. The existing caption of “Significant differences in the fatty acid spectrum.” does not describe the Figure.

Line 221: is there a standard deviation for Figure 6B?

Reviewer #2:

Remarks to the Author:

The article relies on the development of a static and dynamic spheroid cultures derived from bovine mesenchymal cells as a fat component of cultured meat. The authors also explored the 3d bioprinting using an edible hydrogel. The study was well-conducted and is relevant in the field, however some points need to be clarified.

Minor revisions

- “The number of isolated cells was highly donor dependent and varied within the subcutaneous fat for the different body regions (data not shown)”. These data should be shown as a table in supplementary data section.

- “While all samples contained mostly saturated fatty acids (SFA) the proportion of saturated, monounsaturated and polyunsaturated differs considerably”. These data

should be shown as a table in supplementary data section.

- It would be interesting to add the production cost per mg of fat.

Major revisions

- What do the authors mean by single step differentiation protocol?

- Please, clarify the positive control for death immunofluorescence analysis.

- The number of bioprinted spheroids per mL should be discriminated or how the authors reached the cell density value described in methods section.

- In figure 5B, the authors should add phase contrast images to the assessment of spheroid distribution in bioprinted hydrogel

- In figure 6A, it is not clear enough, if the image represents the lipids extracted from spheroids or spheroids. It seems only lipid content.

Reviewer #3:

Remarks to the Author:

The study of Klatt / Kluger et al. presents an interesting and well described process for the production of cultured bovine fat using bASC spheroids. Although the used methods are generally known, experimental work on formation and culture of spheroids using cells from farm animals has been rarely done. However, knowledge and protocols for 3D-cell culture are essentially needed in the field of cultured meat production. Such knowledge is also interesting for stem cell research and the development of stem cell-based therapies. The authors were able to produce bASC spheroids that were stable and viable in static but more importantly in dynamic culture conditions which is preferable for cultured meat production. Dynamic culture of bASC spheroids has been newly described in the current work. The fatty acid profile of bASC spheroids was similar to those of bovine fat but differentiated bASC spheroids contained more MUFA which are healthier. The authors also show that those bASC spheroids can be bioprinted in edible bioinks (here GG) and thereby used to form more complex, fat containing constructs.

In the opinion of the reviewer, the work of Klatt / Kluger et al. makes a significant contribution to the development of scalable processes for the production of cultivated fat.

All experimental procedures and methods are well described and detailed additional data information is given allowing the reproduction or application of the described process by others.

In the following some questions and minor points are given that require clarification/improvement:

Abstract

Line 14

Please use another expression than “lab grown”, e.g., “in vitro” or “using biotechnological methods”. “Lab” is too colloquial and also implies that only a small amount can be produced.

Line 15

“Reduce antimicrobials”: I think at the end no antibiotics should be used! Something like “products will be free of antibiotics” will fit better.

Introduction

Line 30

a) I am not sure if there is really a “research field” because at the moment most research is done in the Startups themselves.

b) See also my comment in line 14 regarding “in the laboratory”.

Lines 41-45

In my opinion more details on regulation of adipogenesis and its regulation are needed. What supplements are typically used in vitro to induce adipogenesis and why? What mechanisms of action do they have? This information is needed for the reader to understand the results.

Results

Lines 65-70, Figure 1A

In my opinion, figure 1A and the related text is not a result and in this context too comprehensive. It should be deleted or relevant parts could be integrated in Figure 1b. Fig. 1b could be rather a good summary.

Line 82

Add “enzymatic” before “isolation”.

Line 91

“...while maintaining a high differentiation rate.” How “a high differentiation rate” is defined? Is “differentiation efficiency” meant?

Line 101

“increase” instead of “influence”

Line 103

Why “only three”? Most classical cocktails use three main supplements. This is why a better explanation of in vitro adipogenic differentiation is needed in the "Introduction" (see Lines 41-45, Introduction)

Lines 115-116

“...using the liquid overlay method.” I am not sure if all readers are familiar with methods for spheroid formation. Give a short explanation.

Line 140

Change “are” to “were”.

Lines 144-145

Irregular size of spheroids can lead to negative effects such as differences in the intra-spheroidal environment resulting in lower differentiation efficiency. How can this be

prevented?

Discussion

Line 229

As mentioned already, there should be more information on the components used in the adipogenic cocktail.

Line 236

Please explain “sustainability” in this context. I see it more in relation to cost reduction and reduction of potentially critical substances. I also would not mix up antibiotics and the differentiation cocktail.

Line 241

I understand that the spheroid size decreased in static and dynamic cultures. Indeed this behavior is seen with MSC spheroids but why? In addition, there are also growing spheroids. Have you ever checked for the existence of growing cells or cells expressing stem cell markers, e.g. in younger spheroides?

Lines 259/260

As the majority of groups work with single cells, the advantages of spheroids should be made more clear in the discussion. Why are spheroids the better option?

Lines 290/291

This is repeated (too) many times in the manuscript.

Methods

Lines 351 and 363: “Afterwards” instead of “Afterward”

Reviewer #1 (Remarks to the Author):

Major comments:

The growth and adipogenic differentiation of bovine adipose-derived stem cells in spheroids under dynamic conditions is an emerging topic for academic and industry players in the cultivated meat field.. The authors have done a good job of performing experiments ranging from cell proliferation, adipogenic differentiation, spheroid culture, dynamic culture, 3D printing, and nutritional profiling. However, many of the results have been redundant from previous publications; for example, spheroid culture (including bovine ASCs; many using human ASCs), reduced adipogenic cocktail, fatty acid composition, and 3D printing/construction of adipocytes. It is thus questioned that the manuscript would significantly advance the technological aspect of cultured fat research. The reviewer feels that it fits better for more specialized journal. Some of the comments on the contents are described below; there are many 'minor' comments for their future reference as the writing needs further substantial improvement.

Thank you for your feedback. While human ASCs are quite well described, information about animal cells, especially fat and fat precursor cells, is scarcely published. The techniques established by us for the dynamic culture and differentiation of bASC spheroids, as well as their use as components in edible bio-inks and subsequent printing, are considered very helpful for the research field in our opinion. We are very thankful for your comments. They helped us to further improve and strengthen our work.

1. Many parts of the article are not clearly explained. It is very hard for the reader to understand.

We have revised the document and have adjusted it for an improved readability.

2. Apart from showing Bodipy and perilipin 1 expression, authors can also supplement with the expression of other adipogenic markers such as PPAR γ and FAS. It can be intracellular staining, western blot or qPCR.

Thank you for your suggestion! To further strengthen our proof of adipogenic differentiation in our spheroids, we decided to include pictures of PPAR γ immunofluorescence stainings. On day 14 of culture PPAR γ was present in the differentiated spheroids but not in the control group. We included those pictures in Figure 3 and 4.

3. The author state that rosiglitazone showed higher bodipy staining as compared to indomethacin (Figure 2D). However, there is no difference in fluorescence intensity per cell (Supplementary Figure 1B). Is there any issue with the image analysis or a result of huge variation? The data from Supp Figure 1B does not look so convincing.

We have a considerable variation between the donors. The cells of different donors react differently to differentiation factors. There is no significant difference between the differentiation results of rosiglitazone and indomethacin. This is the result of those variations. However, we see a trend of rosiglitazone being more reliable to yield a good differentiation efficiency.

4. The authors claim the dynamic culture is novel, but I am not sure if the experiment is convincing

enough. They should have sufficient data to show why it works better than the static culture. Any quantifiable data to show the trend over time?

In fact, we are the first to show a dynamic spheroid culture of bASCs. We are working on implementing a dynamic spheroid culture in a bioreactor setting. This is an important step towards large-scale production of cell mass. With this publication we want to show that spheroid culture with bASCs is possible, and we are able to show differentiation and lipid accumulation. In the future, further tests will be conducted within our working group to analyze the full potential of bASC spheroids in a dynamic system.

5. If the reduction in size of bASC spheroid is not caused by cell death, what caused it?

This is a very interesting aspect. MSC spheroids are in fact known to shrink in culture. With our cell death staining we were able to tell that no cell death could be found within the spheroids on day 14, especially the core. We think there are two main aspects to be considered. Firstly, the spheroids could get more compact due to stronger cell-cell connections built over time. Secondly, cells on the outside of the spheroid could leave the spheroid and form new spheroids or stay as a single cell or even die. We observed in fact a small number of single cells next to the spheroids, mainly during the first three media changes.

6. For nutritional profiling, what is the absolute concentration of the major fatty acids? The cells may have similar composition (e.g. 60-80% SFA), but differ in absolute amount (e.g. 0.2g of SFA per g of cells). This is very important as it has implication on feasibility of this entire technology.

This is an important point. However, in our present studies we focused on fatty acid composition, the total amount of lipids was not determined. In future experiments we will determine the absolute lipid amount.

Minor comments

Line 1: encourage the author to use “cultivated meat” instead of “cultured meat” as an industry norm.

Thank you for your suggestion. We introduced the term cultivated meat equal to cultured meat in our paper.

Line 30: Suggest to remove “without the need to raise the whole animal”. It might be benefit of cultivated meat. However, it does not add any value towards the main message of growing cultivated meat research field.

We have removed this part from the introduction.

Line 33: In future, it could contribute to the nutrition of the [remove the] growing world population while producing less [lesser] greenhouse [gas] emissions and using less [lesser] land than conventional animal husbandry.

We have rephrased this sentence.

Line 38: what are the “hurdles in all parts of the process”?

We added a short description of the most important hurdles and challenges in the introduction.

Line 45: “Usually, adherent cells ... making it time, work, and material consuming” This sentence is too long and contains too many idea. Please revise it.

We revised this part for a better readability.

Line 48: “When employing scaffolds such as non-edible microcarriers, an additional step is necessary to harvest the cells before proceeding with further processing”. Change to “When employing non-edible scaffolds such as microcarriers, an additional step is necessary to harvest the cells before proceeding with further processing”. There are scaffolds that are edible and do not require harvesting or further processing. Thus it is important to be specific to non-edible scaffold.

We changed this sentence for more clarity.

Line 52: What is “so-called spheroids”? If they are spheroids, just term the as spheroid. If they are not, please refrain from using the term to confuse readers.

We removed the “so-called”.

Line 58: “However, very little research has been conducted concerning bovine fat spheroids”. Do not have sufficient link to the previous sentences. It might be meaningful to explain the importance of fat spheroid culture or research done on spheroid from other animal or tissue origin?

We have changed this part to achieve better readability. Sadly, the literature of animal fat spheroids is scarce in general

Line 65: “The development of a new dynamic spheroid culture of bASCs is a delicate process” to change to “The development of a new dynamic spheroid culture of bASCs for cultivated meat is a delicate process”. Authors need to highlight cultivated meat here as such spheroid culture can be used for other purposes that do not need to address challenges of cultivated meat research and challenges.

You are right. Thank you for this suggestion! We changed this sentence accordingly.

Figure 1: font used in the figure looks very different. I would suggest to change to font that are easier to read.

We adjusted the font in Fig. 1 to calibri.

Figure 1 caption: the following sentences are incomplete. Please rephrase them “Technological,

financial, consumer, and regulatory gaps that need to be addressed for cultured meat commercialization” and “Following, characterization, bioprinting and fatty acid analysis.”

We changed the sentences in the caption of Figure 1.

Line 84: change “varied within the 84 subcutaneous fat for the” to “varied between the 84 subcutaneous fat from the”

Thank you for pointing that out! We applied the change in that sentence.

Line 92 – 104: it is very difficult to understand or follow the author. I will suggest to rephrase them. Also, it appears that indomethacin and IBMX might not be needed for adipogenic differentiation. However, did the author investigate if dexamethasone and insulin are needed?

We adjusted this part for better readability. We investigated if insulin and dexamethasone are needed. In our setting we determined that dexamethasone is essential for the differentiation process. Without insulin some differentiation could be found but the efficiency dropped noticeably.

Line 113: What caused the difference in spheroid size after 5 days? The author showed that the expression of cCas3, RIPK3 and Hif-a were absent on Day 14. Did the author investigate look at the expression of cCas3, RIPK3 and Hif-a at different timepoint? What is the message or conclusion that the author is trying to show?

We show the absence of cell death markers on day 14. We want to show that the spheroids did not have a dead/necrotic core. Within the first days of the spheroid culture the spheroids did lose cells and seem to tighten up. This ‘shrinking’ is known to happen in MSC spheroid culture. We don’t know the exact reason and think it's worth investigating this aspect in more detail in the future.

Line 141: “The size distribution was most uniform when maintaining a culture density of 250,000 cells/mL at a shaking speed of 70 rpm during spheroid formation” How did the author measure the size distribution?

We tested different cell densities and shaking speeds and evaluated the results by image analyses. The difference was visually assessed, and we found obvious differences. In general, lower speeds and higher cell counts lead to a mixture of big spheroids (>500µm) and single cells and higher shaking speeds und lower seeding density led to a mixture of spheroids of a size between 50 µm and 200 µm. Speeds of more than 70 rpm negatively influenced cell survival.

Line 143: “For spheroid maintenance and differentiation, a shaking 143 speed of 80 rpm yielded the most favorable results” What did the author mean by most favorable results? Size distribution, cell yield, differentiation efficiency, lipid uptake? What is the range of shaking speed that the author investigated?

In this case ‘most favorable results’ refers to the size distribution of the spheroids and the cell survival. The cells were a little less sensitive than in the spheroid formation phase. We tested shaking speeds between 60-100 rpm.

Line 147: $3.68 \text{ mg} \pm 1.9\text{mg}$ has a huge deviation. Is there any way to reduce the deviation?

We see these huge variations between cells of different donors. Reducing this variation will hopefully be achieved due to a uniform cell source.

Figure 4A: In Day 14, control and differentiated spheroids look similar in size. Why is it that differentiated spheroids look 2x larger than control spheroids in Figure 4C and 4D?

In Figure 4C and D we show cryosections of different spheroids. Depending on the section the diameter of the spheroid is not fully portrayed.

Figure 5C and 5D: the images look out of focus. Will z-step or confocal imaging be meaningful?

In those pictures a z-stack is already applied. As we look at spheroids within a gel of considerable thickness it is complicated to get a good shot. Confocal imaging could be helpful.

Line 195: "While all samples contained mostly saturated fatty acids (SFA) the proportion of saturated, monounsaturated and polyunsaturated differs considerably (Figure 6 B)." to change to "While all samples contained mostly saturated fatty acids (SFA), the proportion of saturated, monounsaturated and polyunsaturated fatty acids differs considerably (Figure 6 B)."

Thank you for your suggestion! We changed the sentence accordingly.

Line 196 – 202: it is very difficult to follow. Please kindly rephrase to help the readers to understand easier.

We changed the part and improved the readability.

Figure 6B and 6C: both figures are showing fatty acid composition with different representations. Please amend the figure caption for 6C.

We rephrased the caption of figure 6C.

Figure 6D and 6E: same as above. The existing caption of "Significant differences in the fatty acid spectrum." does not describe the Figure.

We rephrased the figure description. While figure 6D shows the full fatty acid spectrum that was detected, figure 6E shows only the most abundant fatty acids of the samples.

Line 221: is there a standard deviation for Figure 6B?

Figure 6B is intended to give a quick and rough overview of the fatty acid composition of the samples. Therefore, we do not show standard deviation. We corrected the figure legend.

Reviewer #2 (Remarks to the Author):

The article relies on the development of a static and dynamic spheroid cultures derived from bovine mesenchymal cells as a fat component of cultured meat. The authors also explored the 3d bioprinting using an edible hydrogel. The study was well-conducted and is relevant in the field, however some points need to be clarified.

Thank you for your feedback and the great comments. We improved our manuscript and made it more precise.

Major revisions

- What do the authors mean by single step differentiation protocol?

In a single step differentiation protocol, the differentiation factors are not changed over the differentiation process. We are using the same differentiation factors in the same concentration within the whole differentiation process.

- Please, clarify the positive control for death immunofluorescence analysis.

We used a 'giant' spheroid as control. The 3 days old spheroid with a diameter of over 800 μm was cryosectioned and stained as described in our methods section and used as a positive control for cCas3 and RIPK3. Here we found signals from both antibodies in the core of the spheroid. For Hif1 α we treated the cells with 50 μM of cobalt sulfate for 7 h prior staining as described in the methods section. A clear signal of Hif1 α was found.

- The number of bioprinted spheroids per mL should be discriminated or how the authors reached the cell density value described in methods section.

The cell density was determined from the cells put in the spheroid culture. As the spheroids vary in size, we did not take a spheroid count. We added this detail to the methods section.

- In figure 5B, the authors should add phase contrast images to the assessment of spheroid distribution in bioprinted hydrogel

We have made some phase contrast images of the gels containing spheroids. Since only small parts of the gel are visible per picture, they don't provide a good overview of the spheroid distribution.

- In figure 6A, it is not clear enough, if the image represents the lipids extracted from spheroids or spheroids. It seems only lipid content.

In Figure 6A we are showing our differentiated spheroids in a cell strainer. We can understand the confusion and changed the caption/description.

Minor revisions

- “The number of isolated cells was highly donor dependent and varied within the subcutaneous fat for the different body regions (data not shown)”. These data should be show as a table in supplementary data section.

We are very happy about your interest in this data and added this information in the supplements.

- “While all samples contained mostly saturated fatty acids (SFA) the proportion of saturated, monounsaturated and polyunsaturated differs considerably”. These data should be show as a table in supplementary data section.

We added this information in the supplement.

- It would be interesting to add the production cost per mg of fat.

This is indeed very interesting. Right now, we are demonstrating the possibility of bASC spheroids in a dynamic system. As our system is still within its first steps, we did not consider calculating the production costs.

Reviewer #3 (Remarks to the Author):

The study of Klatt / Kluger et al. presents an interesting and well described process for the production of cultured bovine fat using bASC spheroids. Although the used methods are generally known, experimental work on formation and culture of spheroids using cells from farm animals has been rarely done. However, knowledge and protocols for 3D-cell culture are essentially needed in the field of cultured meat production. Such knowledge is also interesting for stem cell research and the development of stem cell-based therapies. The authors were able to produce bASC spheroids that were stable and viable in static but more importantly in dynamic culture conditions which is preferable for cultured meat production. Dynamic culture of bASC spheroids has been newly described in the current work. The fatty acid profile of bASC spheroids was similar to those of bovine fat but differentiated bASC spheroids contained more MUFA which are healthier. The authors also show that those bASC spheroids can be bioprinted in edible bioinks (here GG) and thereby used to form more complex, fat containing constructs.

In the opinion of the reviewer, the work of Klatt / Kluger et al. makes a significant contribution to the development of scalable processes for the production of cultivated fat.

All experimental procedures and methods are well described and detailed additional data information is given allowing the reproduction or application of the described process by others.

In the following some questions and minor points are given that require clarification/improvement:

We are very thankful for your feedback! We revised the manuscript with the help of your comments.

Abstract

Line

14

Please use another expression than “lab grown”, e.g., “in vitro” or “using biotechnological methods”. “Lab” is too colloquial and also implies that only a small amount can be produced.

We do understand the implications of ‘lab-grown’ and changed it to ‘in vitro’.

Line 15

“Reduce antimicrobials”: I think at the end no antibiotics should be used! Something like “products will be free of antibiotics” will fit better.

We agree. We changed this part to ‘no antimicrobials at all’.

Introduction

Line 30

a) I am not sure if there is really a “research field” because at the moment most research is done in

the Startups themselves.

b) See also my comment in line 14 regarding “in the laboratory”.

a) We see the strong involvement of startups in cultured meat research. Nevertheless, research groups worldwide are working on the development and the deeper understanding of the underlying mechanisms of different aspects of cultured meat research. Industry and academia contribute their results to this field of research which makes it a fast growing and evolving research topic.

b) We changed it to ‘in vitro’

Lines 41-45

In my opinion more details on regulation of adipogenesis and its regulation are needed. What supplements are typically used in vitro to induce adipogenesis and why? What mechanisms of action do they have? This information is needed for the reader to understand the results.

Thank you for your comment, we included some of the most common supplements and their mechanisms of action in the introduction.

Results

Lines 65-70, Figure 1A

In my opinion, figure 1A and the related text is not a result and in this context too comprehensive. It should be deleted or relevant parts could be integrated in Figure 1b. Fig. 1b could be rather a good summary.

We understand your concern. Since nature communications has a broad readership, we wanted to use Fig. 1A to give a brief overview of the diverse challenges in CM. Figure 1B shows a graphical summary of our work presented in the paper. The individual work steps in Figure 1B have numbers that can be found in Figure 1A and are intended to make it clear which challenges we are explicitly addressing. For this reason, we would like to retain Figure 1A and hope for approval.

Line 82

Add “enzymatic” before “isolation”.

Thank you for this comment! We added the word ‘enzymatic’.

Line 91

“...while maintaining a high differentiation rate.” How “a high differentiation rate” is defined? Is “differentiation efficiency” meant?

We changed ‘differentiation rate’ to differentiation efficiency.

Line 101

“increase” instead of “influence”

We changed it accordingly.

Line 103

Why “only three”? Most classical cocktails use three main supplements. This is why a better explanation of in vitro adipogenic differentiation is needed in the "Introduction" (see Lines 41-45, Introduction)

We added a short explanation of in vitro differentiation in the introduction. In the human system many protocols use 3 main supplements or more. In bovine adipogenic differentiation often, a minimum of 4 differentiation supplements is used. Since there is not as much knowledge about the adipogenesis in ruminants compared to humans, lots of groups use more supplement combinations with more supplements in them to ensure a stable differentiation efficiency.

Lines 115-116

“...using the liquid overlay method.” I am not sure if all readers are familiar with methods for spheroid formation. Give a short explanation.

We added a short explanation of the liquid overlay technique.

Line 140

Change “are” to “were”.

Good catch! We changed it accordingly.

Lines 144-145

Irregular size of spheroids can lead to negative effects such as differences in the intra-spheroidal environment resulting in lower differentiation efficiency. How can this be prevented?

In our dynamic system the spheroid size can be modulated through the shaking speed and the cell density. All uniform spheroids are not possible with this method. In static culture spheroids of the same size and cell count are commonly used for experiments. For high throughput experiments spheroid plates can be used, allowing the formation of small and uniform spheroids with less effort in microwells. If the spheroids were produced in a static system and then cultured in a dynamic system, the size distribution would be more even. Spheroid formation in a bioreactor could be also more uniform than on an orbital shaker. In our experiments we saw that a range of spheroid sizes showed good conditions for adipogenic differentiation.

Discussion

Line 229

As mentioned already, there should be more information on the components used in the adipogenic cocktail.

We added information to the components in the introduction and the discussion.

Line 236

Please explain “sustainability” in this context. I see it more in relation to cost reduction and reduction of potentially critical substances. I also would not mix up antibiotics and the differentiation cocktail.

You are right. We used sustainability to imply the reduction of harmful substances and the cost reduction of production. We changed this part to be more precise. We also moved the statement concerning the antibiotic free work.

Line 241

I understand that the spheroid size decreased in static and dynamic cultures. Indeed this behavior is seen with MSC spheroids but why? In addition, there are also growing spheroids. Have you ever checked for the existence of growing cells or cells expressing stem cell markers, e.g. in younger spheroids?

From what we understand the size decrease in MSC spheroids could be due to stronger cell-cell connections over time, making the spheroids more compact and due to shedding of cells that were not fully incorporated in the spheroid. We did not check for growing cells in our spheroids. In this study, our focus lays on the differentiation within the dynamic and static spheroid culture. In future studies we want to investigate the proliferation within the spheroids and the influence of different growth factors on this system.

Lines 259/260

As the majority of groups work with single cells, the advantages of spheroids should be made more clear in the discussion. Why are spheroids the better option?

We added a brief part on some of the important advantages of spheroids.

Lines 290/291

This is repeated (too) many times in the manuscript.

We do state this fact 3 times in our paper. It is a very important point for cultured meat manufacturing. We added other advantages of spheroids to give a more balanced perspective. In future experiments we will be studying if the 3D structure has positive effects on the differentiation efficiency in our systems.

Methods

Lines 351 and 363: "Afterwards" instead of "Afterward"

Thank you! We changed it accordingly.

Reviewers' Comments:

Reviewer #1:

Remarks to the Author:

The authored tried to address many of the concerns that were previously shared. There are a few remaining though, which should be addressed or amended.

The major concern is that many figures show representative images rather than attempt to quantify in numbers though most of experiments are described as n=3 replicates. It is not very certain if images shown are representative or simply picked to support their conclusion.

As an example, what are described in Lines 105 to 114 (essentially data in Figure 2) is not convincing enough. I see a few problems and confusions as follows:

- Figure 2d indicate higher Bodipy signal w/o indomethacin than w/ indomethacin, but the text mentions that indomethacin induced differentiation.
- All the comparison in Supplementary Figure 1 shows high variation between samples, and the results do not look conclusive.
- Did the author assess if Dexamthasone and insulin can be removed from the differentiation media like IBMX?
- "Figure 2" in Line 101 should be indicated as Figure 2c.

Reviewer #2:

Remarks to the Author:

Several relevant informations were added to the article, however there are still two remaining issues:

- Figure 5: none of the figures showed spheroids incorporated in bioink. The authors must show a intermediate magnification (for ex. 20 or 40X).
- Figure 6: the macroscopic aspect of the lipid droplets showed in Figure 6A doesn't corresponding to morphological aspect of lipd droplets showed in Figure 4e. Please, clarify.

Reviewer #3:

Remarks to the Author:

This is a revised version of the manuscript from Klatt / Kluger et al., now named "Dynamically cultured, differentiated bovine adipose-derived stem cell spheroids as building blocks for biofabricating cultured fat".

The authors answered almost all questions I asked in my last review. Likewise, most of the requested changes have been made. In particular, the in vitro methods of adipogenic differentiation were better presented. Together with the changes suggested by the additional reviewers, the readability and content of the manuscript has improved significantly.

In my opinion, due to the complex and new topic of cultured fat production, many questions can only be clarified with further experimental work, which, however, goes far beyond the scope of this work.

Therefore, I have only the following minor points requiring clarification/improvement:

Line 105-106: Clearly, there is no significant difference between the protocols with indomethacin and rosiglitazone. Thus, at this time point and with only data from three animals (N = 3), the conclusion that indomethacin “caused less lipid accumulation” is not conclusive. It should therefore be worded more carefully, e.g. “induced visually less lipid accumulation”.

The low number of animals or of slaughter material/tissue from those animals (all from one race) is a general weakness of the work. This means that quantitative statements in particular can only be viewed with great caution.

Legend figure 2: It will be easier for the reader to understand the experiments and this figure if “DMEM-Diff” medium will be explained in the text and figure legend.

Line 143 “xxx, the differentiated bASC spheroids accumulated...”: change “the” to “that”

REVIEWER COMMENTS

Reviewer #1 (Remarks to the Author):

The authors tried to address many of the concerns that were previously shared. There are a few remaining though, which should be addressed or amended.

The major concern is that many figures show representative images rather than attempt to quantify in numbers though most of experiments are described as n=3 replicates. It is not very certain if images shown are representative or simply picked to support their conclusion.

Thank you very much for the opportunity to clarify this uncertainty. The images in the figures were selected to best represent the overall results. To demonstrate to you that our image selection in the manuscript accurately reflects the overall results, we have included here a larger selection of images. Upon revisiting the raw data, we unfortunately noticed that the differentiated spheroid at day 14 in Figure 3e was mistakenly chosen from a 10k differentiated spheroid. We apologize for the mix-up and have accordingly substituted the image with a differentiated 50k spheroid.

Figure 3c

Figure 3d

Figure 3e

Figure 4c

Figure 4d

Figure 4e

Again, to further confirm the validity of our statements, we have now examined the success of the differentiations in both static and dynamic spheroids **by quantifying the intensities of the fat differentiation markers or lipid stains following your suggestion**. The data from the quantifications confirm the success of the differentiation protocol in line with existing images in the figures. The corresponding data are now included in Supplementary Fig. 2 and 3.

Supplement Figure 2d Quantification of static 50k spheroid differentiation

Supplement Figure 3 Quantification of dynamic spheroid differentiation

As an example, what is described in Lines 105 to 114 (essentially data in Figure 2) is not convincing enough. I see a few problems and confusions as follows:

- Figure 2d indicate higher Bodipy signal w/o indomethacin than w/ indomethacin, but the text mentions that indomethacin induced differentiation.

Thank you for this important hint. During the first revision, symbols in Figures 2d and 2g were changed from "+" and "-" to "w and w/o" inadvertently resulting in a mix-up of the symbols. We sincerely apologize for this error and any resulting confusion and have corrected the labels accordingly.

- All the comparison in Supplementary Figure 1 shows high variation between samples, and the results do not look conclusive.

We agree with your concerns. When quantifying the intracellular lipids or lipid intensity per nucleus, we did not normalize the fluorescence signals to the total cell area since we did not use additional counter staining for the cell membrane or other cellular structures. Therefore, we could only normalize the signal intensities to the number of cells or nuclei. If differentiated cells accumulate lipids to varying degrees or possess lipid droplets of different sizes, this normalization may lead to somewhat distorted results despite the visible overall trend of the differentiation success.

We were aware that this type of quantification would lead to certain variations. However, when quantifying the differentiation success of the spheroids, normalization of fluorescence intensities to the area under investigation is possible, and here the success of the established differentiation protocol became more apparent. The collective findings equally confirm that in the present study, we were able to develop a suitable differentiation medium for our production protocol of building blocks for cultivated meat production

- Did the author assess if Dexamethasone and insulin can be removed from the differentiation media like IBMX?

We acknowledge your concerns regarding the reduction of insulin and dexamethasone. The optimization of our differentiation medium aimed at establishing a well-functioning differentiation process for adipose tissue. In the future, we intend to further optimize this medium towards food-grade standards, thereby facilitating potential approvals in the food industry.

- "Figure 2" in Line 101 should be indicated as Figure 2c.

Thank you. We added the c.

Reviewer #2 (Remarks to the Author):

Several relevant informations were added to the article, however there are still two remaining issues:
- Figure 5: none of the figures showed spheroids incorporated in bioink. The authors must show a intermediate magnification (for ex. 20 or 40X).

We took pictures with 50, 100 and 200x magnification of the spheroids incorporated in gellan gum. Here we observed the spheroids and their distribution within the gel using a 10x magnification, allowing us to determine the expected paths of the gel structures based on the spheroid distribution.

However, in this intermediate magnification, we were unable to make the gellan gel clearly visible. Therefore, we decided not to include this intermediate magnification as it would not provide significant additional information compared to the other micro and macroscopic images in the figures. Instead, we were able to replace the original image with the printed grid including spheroids (Figure 5b) with a higher resolution version, resulting in improved visibility of at least the larger sized spheroids.

- Figure 6: the macroscopic aspect of the lipid droplets showed in Figure 6A doesn't corresponding to morphological aspect of lipid droplets showed in Figure 4e. Please, clarify.

In Figure 6a we show a piece of bovine fat tissue and the cultured fat spheroids in a macroscopic scale. Figure 4e shows cryosections of spheroids with a 200x magnification and stained lipid content of the spheroids. The outer appearance of the harvested spheroids (Fig. 6a) without any magnification is not comparable with the morphology of the lipid droplets inside spheroids (Fig 4e).

Reviewer #3 (Remarks to the Author):

This is a revised version of the manuscript from Klatt / Kluger et al., now named “Dynamically cultured, differentiated bovine adipose-derived stem cell spheroids as building blocks for biofabricating cultured fat”.

The authors answered almost all questions I asked in my last review. Likewise, most of the requested changes have been made. In particular, the in vitro methods of adipogenic differentiation were better presented. Together with the changes suggested by the additional reviewers, the readability and content of the manuscript has improved significantly.

In my opinion, due to the complex and new topic of cultured fat production, many questions can only be clarified with further experimental work, which, however, goes far beyond the scope of this work.

Therefore, I have only the following minor points requiring clarification/improvement:

Line 105-106: Clearly, there is no significant difference between the protocols with indomethacin and rosiglitazone. Thus, at this time point and with only data from three animals (N = 3), the conclusion that indomethacin “caused less lipid accumulation” is not conclusive. It should therefore be worded more carefully, e.g. “induced visually less lipid accumulation”.

Thank you for this suggestion. We changed the sentence accordingly.

The low number of animals or of slaughter material/tissue from those animals (all from one race) is a general weakness of the work. This means that quantitative statements in particular can only be viewed with great caution.

Indeed, the cell source for Cultured Meat poses a significant challenge, as reliable cell sources for various tissue types such as fat and muscle are needed. We are working with adult stem cells, which are utilized by many research groups, but may not be the ideal cell source for future mass production of cultured meat in some respects. Despite the challenges in handling and cultivation associated with this cell source, we have nevertheless been able to demonstrate a proof of concept for the production of building blocks for cultured meat production. With this knowledge, potentially more suitable cell sources such as cell lines or iPSCs can benefit in the future. We emphasized this aspect in the beginning of the discussion.

Legend figure 2: It will be easier for the reader to understand the experiments and this figure if “DMEM-Diff” medium will be explained in the text and figure legend.

Thank you for this suggestion! We added the composition of DMEM-Diff in the figure legend and the text.

Line 143 “xxx, the differentiated bASC spheroids accumulated...”: change “the” to “that”

Thank you for this hint. We changed it accordingly.

Reviewers' Comments:

Reviewer #1:

Remarks to the Author:

In the revised manuscript, the authors have addressed all of my previous concerns and comments.

Reviewer #2:

Remarks to the Author:

I am not convinced with the authors answers.

Related to the issue about the magnification of bioprinted spheroids images, the images showed by the author do not resemble a bioprinted hydrogel; it resembles spheroids incorporated in a hydrogel solution. The replaced image incorporated into the manuscript is even worse - the white spots do not resemble spheroids and the scale bar is missing (important to attest spheroid size).

Related to the macroscopic view of fatty spheroids showed at 6a, the content of lipid doesn't match to the content of intracytoplasmic lipid of spheroids showed in the manuscript.

Reviewer #3:

Remarks to the Author:

All my comments have been addressed.

I have no further comments.

One very small change: Please add spheroides before 5.8% in line 230.

REVIEWER COMMENTS

Reviewer #1 (Remarks to the Author):

In the revised manuscript, the authors have addressed all of my previous concerns and comments.

Reviewer #2 (Remarks to the Author):

I am not convinced with the authors answers.

Related to the issue about the magnification of bioprinted spheroids images, the images showed by the author do not resemble a bioprinted hydrogel; it resembles spheroids incorporated in a hydrogel solution. The replaced image incorporated into the manuscript is even worse - the white spots do not resemble spheroids and the scale bar is missing (important to attest spheroid size).

Thank you very much for your feedback. Especially, the hint about the scale bars is very helpful. Until now, we had only mentioned the size of the constructs in the methods section. We completely agree that the size must be immediately recognizable in Figure 5 as well. Therefore, we have added the full size of the printed grid structure to the figure caption (new text in green). Additionally, we have included scale bars in the photos, which make it easier to attest spheroid size.

For improving the photo quality, we have captured additional images of the hydrogels, both with and without differentiated spheroids incorporated, and updated Figure 5b accordingly. In these new, improved images, we can now clearly see the spheroids. Moreover, we provide new magnified images in the supplements, which further support this statement (new Supplementary Figure 3b, new figure caption text 3b in green). We are convinced that these new images provide a clearer depiction of the printed grid structure and the incorporated spheroids. The hydrogel structure without cells is clearly visible in the image next to the one showing the hydrogel with incorporated spheroids. Furthermore, it is important to note that single cells would not be visible in macroscopic images.

Related to the macroscopic view of fatty spheroids showed at 6a, the content of lipid doesn't match to the content of intracytoplasmic lipid of spheroids showed in the manuscript.

Thank you for your insightful comment. We apologize for this misunderstanding. We would like to clarify that the photo in question (Figure 6a) was intended solely to provide an overview of our harvested spheroids next to native bovine fat tissue. It was not meant to serve as a quantitative analysis of lipid content. The actual characterization of our spheroids, including lipid content, was carried out using various qualitative and quantitative methods, as detailed in the different figures throughout the manuscript. At no point did we intend to draw any quantitative conclusions about the lipid content from the macroscopic image. The quantification of lipid content was performed after lipid staining of accumulated intracellular lipids in cross-sections captured following the cryosectioning of the spheroids (Figure 3 and Supplementary Figure 2d for static and Figure 4 and Supplementary Figure 3a for dynamic spheroid production and differentiation).

As mentioned above the macroscopic images in Figure 6a are intended solely to provide a superficial overview of the spheroids. To enhance clarity for our readers, we have the title of figure 6a to "Macroscopic images of native fat tissue and differentiated spheroids" and the figure caption (in green) to underscore that this image is meant as an overview. We have further updated the corresponding section in our manuscript where we previously introduced the images of fat tissue and differentiated spheroids, ensuring that any unintended comparisons of these structures have been removed (p. 12, l. 217-218, new text in green). Moreover, we changed in Figure 1b the text of step 6 from "Composition close to original" to "fatty acid profile close to native fat". We hope this explanation and the updated manuscript address your concerns.

Reviewer #3 (Remarks to the Author):

All my comments have been addressed.

I have no further comments.

One very small change: Please add spheroids before 5.8% in line 230.

Thank you for this hint, we have added the word "spheroids" in the sentence as suggested.